# On Products of Random Matrices

**DOI:** 10.3390/e22090972

**Published:** 2020-08-31

**Authors:** Natalia Amburg, Aleksander Orlov, Dmitry Vasiliev

**Affiliations:** 1A.I. Alikhanov Institute for Theoretical and Experimental Physics of NRC Kurchatov Institute, B. Cheremushkinskaya, 25, 117259 Moscow, Russia; amburg@itep.ru (N.A.); vasiliev@itep.ru (D.V.); 2Institute for Information Transmission Problems of RAS (Kharkevich Institute), Bolshoy Karetny per. 19, build.1, 127051 Moscow, Russia; 3Moscow Center for Fundamental and Applied Mathematics, 119991 Moscow, Russia; 4Institute of Oceanology, Nahimovskii Prospekt 36, 117997 Moscow, Russia; 5Moscow Institute of Physics and Technology, 141701 Dolgoprudny, Russia

**Keywords:** random complex and random unitary matrices, matrix models, products of random matrices, Schur polynomial, Hurwitz number, generalized hypergeometric functions, integrable systems

## Abstract

We introduce a family of models, which we name matrix models associated with children’s drawings—the so-called dessin d’enfant. Dessins d’enfant are graphs of a special kind drawn on a closed connected orientable surface (in the sky). The vertices of such a graph are small disks that we call stars. We attach random matrices to the edges of the graph and get multimatrix models. Additionally, to the stars we attach source matrices. They play the role of free parameters or model coupling constants. The answers for our integrals are expressed through quantities that we call the “spectrum of stars”. The answers may also include some combinatorial numbers, such as Hurwitz numbers or characters from group representation theory.

## 1. Introduction

Interest in matrix integrals arose in different contexts and at different times. These are problems of statistics in biology (Wishart), and problems of quantum chaos (Wigner, Dyson, Gorkov, Eliashberg, Efetov), and purely mathematical problems of representation theory (see the textbook [1]). At the end of the previous century, applications were added in the theory of elementary particles (t’Hooft [2]), statistical physics (Migdal, Itsikson and Zuber [3], and Kazakov and Kostov [4]) and string theory (Kazakov and Brezin [5], and Migdal and Gross [6]). Now new applications have been added: in the theory of information transfer and theory of quantum information ([7] or lectures [8] (Ch. 10 on quantum Shannon theory)). We allow ourselves not to provide an ocean of links for each of these areas.

Moving closer to the point, we only refer to works on the use of random matrices in [9], works about products of complex random matrices [10,11,12], a review [13,14,15,16] and works close to ours from a mathematical point of view [17,18,19,20,21,22].

In the theory of information, the product of the matrices describes the cascade transformation of signals, and averaging over the matrices means introducing interference and noise. The task is to calculate various correlation functions in such models.

We offer a family of models that in a sense can be called exactly solvable (cf. [20]). They are built according to the so-called children’s drawings, more precisely, clean children’s drawings, which in combinatorics are also called maps. This is a graph drawn on a closed orientable surface, which has the following property: if we cut it along the edges, the surface decomposes into regions homeomorphic to disks (that is, they can be turned into disks by continuous transformation). In addition to this picture, we turn the vertices of the graph into small disks; we call these disks stars.

The edges of the graph are assigned matrices over which the integration is performed. Depending on the graph we draw, we will get one model or another.

Surprisingly, it turns out that studying the products of random matrices with sources is an easier and more natural task. Writing answers for such integrals turns out to be a faster task if we have these matrices. The absence of these matrices is equivalent to some additional averaging that needs to be specifically monitored.

Writing out some answers requires knowledge of certain combinatorial numbers for which tables exist. In one case, these are the so-called Hurwitz numbers; in the other, these are the characters of representations of a symmetric group. However, sometimes the answers are simplified and written in a fairly simple form-in the form of determinants or in the form of products.

Adding source matrices leads to a variety of interesting relationships with differential operators. We briefly mention this in the Appendix E.

## 2. Technical Tools

### 2.1. Partitions; Power Sums and Schur Functions; Hurwitz Numbers

Here we follow [23]. Further technical details are in Appendix A and Appendix B

Partitions. The *partition*
$\lambda =(\lambda_{1},\ldots ,\lambda_{k})$ is a set of nonnegative integers $\lambda_{i}$ which are called parts of λ and which are ordered as $\lambda_{i}\geq \lambda_{i+1}$. The number of non-vanishing parts of λ is called the length of the partition λ, and will be denoted by ℓ(λ). The number $\left|\lambda \right|=\sum_{i}\lambda_{i}$ is called the weight of λ. The set of all partitions will be denoted by Y and the set of partitions of weight *d* by $Y_{d}$. Example: the partition (4,4,1) belongs to $Y_{9}$. The length of (4,4,1) is 3.

We shall use Greek letters for partitions.

Sometimes, it is convenient to use a different notation
$$\lambda =\left(1^{m_{1}}2^{m_{2}}3^{m_{3}}\cdots \right),$$
where $m_{k}$ is the number of times an integer *k* occurs as a part of λ. For instance, (4,4,1) may be written as $\left(1^{1}4^{2}\right)$. The number
(1)$$z_{\lambda }=\prod_{k>0}k^{m_{k}}m_{k}!$$
plays an important role hereinafter.

Example: $z_{\left(4,4,1\right)}=z_{\left(1^{1}4^{2}\right)}=1^{1}1!4^{2}2!=32$.

Partitions can be perceived visually using Young diagrams (YDs): a partition λ with parts $\lambda_{1},\ldots ,\lambda_{\ell }$ matches the rows of the corresponding YD of the length $\lambda_{1},\ldots ,\lambda_{\ell }$, respectively; see [23] for details. The weight of λ is the area of the YD of λ.

Power sums. For a matrix $X\in GL_{N}\left(C\right)$, we put
(2)$$p_{k}\left(X\right):=\mathrm{tr}X^{k}=x_{1}^{k}+\cdots +x_{N}^{k},\;k=1,2,\ldots ,$$
which are the Newton sums of its eigenvalues.

Then, for a partition $\Delta =(\Delta_{1},\ldots ,\Delta_{k})$, we introduce
(3)$$p_{\Delta }\left(X\right)=\mathrm{tr}\left(X^{\Delta_{1}}\right)\ldots \mathrm{tr}\left(X^{\Delta_{\ell }}\right).$$

We put $p_{\left(0\right)}\left(X\right)=1$.

**Remark** **1.**
*Let us note that $p_{\Delta }\left(X\right)$ is a polynomial in the eigenvalues of X and also a polynomial in entries of X. We consider $p_{\Delta }\left(X\right)$ to be a map $GL_{N}\left(C\right)\to C$.*


The polynomial $p_{\Delta }$ is a symmetric function of the eigenvalues $x_{1},\ldots ,x_{N}$ and called the *power sum* labeled with a multi-index Δ.

If one assigns a degree 1 to each $x_{i}$, then the degree of $p_{\Delta }\left(X\right)$ is |Δ|.

In many problems, the set $p_{1},p_{2},\ldots$ is considered as an independent set of complex parameters, which are in no way associated with any matrix. In this case, instead of p(X) we write simply $p=(p_{1},p_{2},\ldots )$. The degree of $p_{m}$ is equal to *m*.

Schur functions. Next, let us recall the definition of the *Schur polynomial*, parametrised by a partition. Consider the equality
$$e^{\sum_{m>0}\frac{1}{m}z^{m}p_{m}}=1+zp_{1}+z^{2}\frac{p_{1}^{2}+p_{2}}{2}+\cdots =\sum_{m\geq 0}z^{m}s_{\left(m\right)}\left(p\right).$$

The polynomials $s_{\left(m\right)}$ are called elementary Schur functions. As we can see, $s_{\left(0\right)}\left(p\right)=1$. Let $\lambda =[\lambda_{1},\lambda_{2},\ldots ]$ be a partition, and define the polynomial $s_{\lambda }$ by
(4)$$s_{\lambda }\left(p\right)=\det{\left(s_{\left(\lambda_{i}-i+j\right)}\left(p\right)\right)}_{i,j\geq 1}.$$

(On the right-hand side, it is assumed that $s_{\left(m\right)}=0$ for negative *m*.) Here we define the Schur function as a function of the set of free parameters $p=(p_{1},p_{2},\ldots )$.

Let us remember that we chose the variables $p=(p_{1},p_{2},\ldots )$ to be related to a matrix $X\in {\mathrm{GL}}_{N}\left(C\right)$. Then $s_{\lambda }$ is a map ${\mathrm{GL}}_{N}\left(C\right)\to C$; we will write
$$s_{\lambda }\left(X\right):=s_{\lambda }\left(p\left(X\right)\right).$$

**Remark** **2.**
*Let us note that $s_{\lambda }\left(X\right)$ is a polynomial in entries of X and a symmetric polynomial in the eigenvalues $x_{1},\ldots ,x_{N}$ of degree d=|λ|.*


**Remark** **3**
**([23]).**
*The Schur functions $s_{\lambda }\left(x_{1},\ldots ,x_{N}\right)$, where ℓ(λ)≥N, form a Z-basis of the space of symmetric polynomials in $x_{1},\ldots ,x_{N}$ of degree d.*


In terms of the eigenvalues $x_{1},\ldots ,x_{N}$, the Schur function reads as
(5)$$s_{\lambda }\left(X\right)=\frac{\det{\left[x_{j}^{\lambda_{i}-i+N}\right]}_{i,j\leq N}}{\det{\left[x_{j}^{-i+N}\right]}_{i,j\leq N}}$$
for ℓ(λ)≤N, and vanishes for ℓ(λ)>N. One can see that $s_{\lambda }\left(x\right)$ is a symmetric homogeneous polynomial of degree |λ| in the variables $x_{1},\ldots ,x_{N}$.

**Remark** **4.**
*The polynomial $s_{\lambda }$ is the character of the irreducible representation of the ${\mathrm{GL}}_{N}\left(C\right)$ group labeled by λ.*


Character map relation. Relation (Equation 4) relates polynomials $s_{\lambda }$ and $p_{\Delta }$ of the same degree d=|λ|=|Δ|. Explicitly, one can write
(6)$$p_{\Delta }=\sum_{\lambda \in Y_{d}}\frac{\dim\lambda }{d!}z_{\Delta }\varphi_{\lambda }\left(\Delta \right)s_{\lambda }\left(p\right)$$
and
(7)$$s_{\lambda }\left(p\right)=\frac{\dim\lambda }{d!}\sum_{\Delta \in Y_{d}}\varphi_{\lambda }\left(\Delta \right)p_{\Delta }.$$

The last relation is called the character map relation. Here
(8)$$\frac{\dim\lambda }{d!}:=\frac{\prod_{i<j\leq N}^{}\left(\lambda_{i}-\lambda_{j}-i+j\right)}{\prod_{i=1}^{N}\left(\lambda_{i}-i+N\right)!}=s_{\lambda }\left(p_{\infty }\right)$$
(see Example 1 in Section 1 and Example 5 in Section 3 of Chapter I in [23]), where
(9)$$p_{\infty }=\left(1,0,0,\ldots \right).$$
and where N≥ℓ(λ). As one can check, the right-hand side does not depend on *N*. (We recall that $\lambda_{i}=0$ in case i>ℓ(λ)). The number dimλ is an integer.

The factors $\varphi_{\lambda }\left(\Delta \right)$ satisfy the following orthogonality relations.
(10)$$\sum_{\lambda \in Y_{d}}{\left(\frac{\dim\lambda }{d!}\right)}^{2}\varphi_{\lambda }\left(\mu \right)\varphi_{\lambda }\left(\Delta \right)=\frac{\delta_{\Delta ,\mu }}{z_{\Delta }}$$
and
(11)$${\left(\frac{\dim\lambda }{d!}\right)}^{2}\sum_{\Delta \in Y_{d}}z_{\Delta }\varphi_{\lambda }\left(\Delta \right)\varphi_{\mu }\left(\Delta \right)=\delta_{\lambda ,\mu }.$$

**Remark** **5.**
*Equality (Equation 7) expresses the ${\mathrm{GL}}_{N}$ character $s_{\lambda }$ in terms of characters $\chi_{\lambda }$ of the symmetric group $S_{d}$ labeled by the same partition λ and evaluated on cycle classes $\Delta \in Y_{d}$. This explains the name of (Equation 7). The numbers $\varphi_{\lambda }\left(\Delta \right)$ are called the normalized characters:*
$$\chi_{\lambda }\left(\Delta \right)=\frac{\dim\lambda }{d!}z_{\Delta }\varphi_{\lambda }\left(\Delta \right).$$

*The integer dimλ is the dimension of the representation λ; that is, $\dim\lambda =\chi_{\lambda }\left((1^{d})\right)$. Equations (Equation 10) and (Equation 11) are the orthogonality relation for the characters.*


Hypergeometric tau functions and determinantal formulas.

Here we follow [24,25]. Let *r* be a function on the lattice Z. Consider the following series
(12)$$1+r\left(n+1\right)x+r\left(n+1\right)r\left(n+2\right)x^{2}+r\left(n+1\right)r\left(n+2\right)r\left(n+3\right)x^{3}+\cdots =:\tau_{r}\left(n,x\right)$$

Here *n* is an arbitrary integer. This is just a Taylor series for a function $\tau_{r}$ with a given *n* written in a form that is as close as possible to typical hypergeometric series. Here *n* is just a parameter that will be of use later. If as *r* we take a rational (or trigonometric) function, we get a generalized (or basic) hypergeometric series. Indeed, take
(13)$$r\left(n\right)=\frac{\prod_{i=1}^{p}\left(a_{i}+n\right)}{n\prod_{i=1}^{q}\left(b_{i}+n\right)}$$

We obtain
τr(n,x)=∑m≥0∏i=1p(ai+n)mm!∏i=1q(bi+n)mxm=pFqa1+n,…,ap+nb1+n,…,bq+n∣x
where
(14)$${\left(a\right)}_{n}=a\left(a+1\right)\cdots \left(a+n-1\right)=\frac{\Gamma (a+n)}{\Gamma (a)}$$
is the Pochhammer symbol.

One can prove the following formula which express a certain series over partitions in terms of (Equation 12) (see, for instance, [24]):(15)$$\tau_{r}\left(n,X,Y\right):=\sum_{\lambda }r_{\lambda }\left(n\right)s_{\lambda }\left(X\right)s_{\lambda }\left(Y\right)=\frac{c_{n}}{c_{n-N}}\frac{\det{\left[\tau_{r}\left(n-N+1,x_{i}y_{j}\right)\right]}_{i,j\leq N}}{\det{\left[x_{i}^{N-k}\right]}_{i,k\leq N}\det{\left[y_{i}^{N-k}\right]}_{i.k\leq N}}$$
where $c_{k}=\prod_{i=0}^{k-1}{\left(r(i)\right)}^{i-k}$ and where
(16)$$r_{\lambda }\left(n\right):=\prod_{(i,j)\in \lambda }r\left(n+j-i\right),\;r_{\left(0\right)}\left(n\right)=1$$
where the product ranging over all nodes of the Young diagram λ is the so-called content product (which has the meaning of the generalized Pochhammer symbol related to λ). Let us test the formula for the simplest case r≡1:(17)$$\det\left[1-X⊗Y\right]=\sum_{\lambda }s_{\lambda }\left(X\right)s_{\lambda }\left(Y\right)=\frac{\det{\left[{\left(1-x_{i}y_{j}\right)}^{-1}\right]}_{i,j\leq N}}{\det{\left[x_{i}^{N-k}\right]}_{i,k\leq N}\det{\left[y_{i}^{N-k}\right]}_{i.k\leq N}}$$

Sometimes we also use infinite sets of power sums $p^{i}=\left(p_{1}^{\left(i\right)},p_{1}^{\left(i\right)},\ldots \right)$ and instead of matrices *X* and *Y* set:(18)$$\tau_{r}\left(n,p^{1},p^{2}\right):=\sum_{\lambda }r_{\lambda }\left(n\right)s_{\lambda }\left(p^{1}\right)s_{\lambda }\left(p^{2}\right)$$

An example r≡1:(19)$$\tau_{1}\left(n,p^{1},p^{2}\right):=e^{\sum_{m>0}\frac{1}{m}p_{m}^{\left(1\right)}p_{m}^{\left(2\right)}}=\sum_{\lambda }s_{\lambda }\left(p^{1}\right)s_{\lambda }\left(p^{2}\right)$$

In case the function *r* has zeroes, there exists a determinantal representation. Suppose r(0); then $r_{\lambda }\left(n\right)=0$ if ℓ(λ)>n. Then
(20)τr(n,p1,p2)=∑λℓ(λ)≤nrλ(n)sλ(p1)sλ(p2)=cndet∂p1(1)a∂p1(2)bτr(1,p1,p2)a,b=0,…,n−1,
where $c_{k}=\prod_{i=1}^{k-1}{\left(r(i)\right)}^{i-k}$, and where
$$\tau_{r}\left(1,p^{1},p^{2}\right)=1+\sum_{m>0}r\left(1\right)\cdots r\left(m\right)s_{\left(m\right)}\left(p^{1}\right)s_{\left(m\right)}\left(p^{2}\right);$$
see [26].

In addition, there is the following formula:(21)$$\tau_{r}\left(n,X,p\right)=\sum_{\lambda }r_{\lambda }\left(n\right)s_{\lambda }\left(X\right)s_{\lambda }\left(p\right)=\frac{\det{\left[x_{i}^{N-k}\tau_{r}\left(n-k+1,x_{i},p\right)\right]}_{i,k\leq N}}{\det\left[x_{i}^{N-k}\right]}$$
where
(22)$$\tau_{r}\left(n,x_{i},p\right)=1+\sum_{m>0}r\left(n\right)\cdots r\left(n+m\right)x_{i}^{m}s_{\left(m\right)}\left(p\right)$$

Let us test it for r≡1, $p=p_{\infty }:=\left(1,0,0,\ldots \right)$:(23)$$e^{\mathrm{tr}X}=\sum_{\lambda }s_{\lambda }\left(X\right)s_{\lambda }\left(p_{\infty }\right)=\frac{\det{\left[x_{i}^{N-k}e^{x_{i}}\right]}_{i,k\leq N}}{\det\left[x_{i}^{N-k}\right]}$$

There are similar series
$$\sum_{\lambda }r_{\lambda }\left(n\right)s_{\lambda }\left(X\right)$$
which can be written as a Pfaffian [27]; however, we will not use them in the present text.

Some properties of the Schur functions. Let us consider

**Lemma** **1.**
*For $X\in {\mathrm{GL}}_{N}$, where detX≠0, and for $\lambda =(\lambda_{1},\ldots ,\lambda_{N})$, ℓ(λ)≤N, we have*
(24)$$s_{\lambda }\left(X\right)\det\left(X^{\alpha }\right)=s_{\lambda +\alpha }\left(X\right)$$
*where α is a nonnegative integer and where λ+α denotes the partition with parts $\left(\lambda_{1}+\alpha ,\lambda_{2}+\alpha ,\ldots ,\lambda_{N}+\alpha \right)$, in particular*
(25)$$s_{\lambda }\left(I_{N}\right)=s_{\lambda +\alpha }\left(I_{N}\right).$$

*Additionally*
(26)$$\frac{s_{\lambda }\left(p_{\infty }\right)}{s_{\lambda +\alpha }\left(p_{\infty }\right)}=\prod_{i=1}^{N}\frac{\Gamma (h_{i}+1+\alpha )}{\Gamma (h_{i}+1)}=\frac{{\left(N+\alpha \right)}_{\lambda }}{{\left(N\right)}_{\lambda }}$$


Hurwitz number. Let E≤2 be an integer and $\Delta^{1}=\left(\Delta_{1}^{1},\Delta_{2}^{1},\ldots \right),\ldots ,\Delta^{k}=\left(\Delta_{1}^{k},\Delta_{2}^{k},\ldots \right)\in Y_{d}$. One can show that
(27)$$H_{E}\left(\Delta^{1},\ldots ,\Delta^{k}\right)=\sum_{\lambda \in Y_{d}}{\left(\frac{\dim\lambda }{d!}\right)}^{E}\varphi_{\lambda }\left(\Delta^{1}\right)\cdots \varphi_{\lambda }\left(\Delta^{k}\right)$$
is a rational number. This number is called the Hurwitz number, which is a popular combinatorial object in many fields of mathematics (see, for instance, [28,29]) and also in physics [30]. The explanations of the geometrical and combinatorial meanings of Hurwitz numbers may be found in Appendix C and Appendix D. We also need a weighted Hurwitz number
(28)$$H_{E}\left(\Delta^{1},\ldots ,\Delta^{k}|m\right)=\sum_{\lambda \in Y_{d}}{\left(\frac{\dim\lambda }{d!}\right)}^{E}\varphi_{\lambda }\left(\Delta^{1}\right)\cdots \varphi_{\lambda }\left(\Delta^{k}\right){\left(\frac{1}{{\left(N\right)}_{\lambda }}\right)}^{m}$$

### 2.2. Mixed Ensembles of Random Matrices

A complex number $E_{n_{1},n_{2}}\left\{f\right\}$ is the notation of the expectation values of a function *f* which depends on the entries of matrices $Z_{1},\ldots ,Z_{n_{1}}\in {\mathrm{GL}}_{N}\left(C\right)$ and of matrices $U_{1},\ldots ,U_{n_{2}}\in U_{N}$:(29)$$E_{n_{1},n_{2}}\left\{f\right\}=\int f\left(Z_{1},\ldots ,Z_{n_{1}},U_{1},\ldots ,U_{n_{2}}\right)d\Omega_{n_{1},n_{2}},$$
(30)$$d\Omega_{n_{1},n_{2}}=\prod_{i=1}^{n_{1}}d\mu \left(Z_{i}\right)\prod_{i=1}^{n_{2}}d_{*}U_{i},$$
where $d_{*}U_{i}$ ($i=1,\ldots ,n_{2}$) is the Haar measure on $U_{N}$ and where
$$d\mu \left(Z_{i}\right)=c\prod_{a,b}e^{-\hbar^{-1}{\left|{\left(Z_{i}\right)}_{a,b}\right|}^{2}}d^{2}{\left(Z_{i}\right)}_{a,b}$$
is the Gaussian measure. Here *ℏ* is a parameter; usually it is chosen to be $N^{-1}$.

Each set of $Z_{i}$ and $d\mu (Z_{i})$ is called complex Ginibre ensemble and the whole set $Z_{1},\ldots ,Z_{n_{1}}$ and $d\mu \left(Z_{1}\right),\ldots ,d\mu \left(Z_{n_{1}}\right)$ are called $n_{1}$
*independent complex Ginibre ensembles*. The set $U_{1},\ldots ,U_{n_{2}}$ and the measure $d_{*}U_{1},\ldots ,d_{*}U_{n_{2}}$ are called $n_{2}$
*independent circular ensembles*. We assume each $\int d_{*}U_{i}=\int d\mu \left(Z_{i}\right)=1$.

The ensemble of the matrices $Z_{1},\ldots ,Z_{n_{1}},U_{1},\ldots ,U_{n_{2}}$ together with the probability measure $d\Omega_{n_{1},n_{2}}$ (that is, with expectation values defined by (Equation 29)) we call a $\left(n_{1},n_{2}\right)$
*mixed ensemble*. We will consider mixed ensembles with $n=n_{1}+n_{2}$ random matrices.

### 2.3. Integrals of Schur Functions and Integrals of Power Sums

In what follows we study expectation values of Schur functions and of power sums. We need four key lemmas.

These lemmas should be known in parts corresponding to Schur functions, but at the moment I have not found all the lemmas along with the proofs, so I will try to fill this gap.

In the Lemmas 2–5 below *A* and *B* are N×N
*complex* matrices; this point is the key.

Everywhere in this section, d=1,…,N.

**Lemma** **2.**
*(I) For any $\lambda \in Y_{d}$ we have*
(31)$$E_{1,0}\left\{s_{\lambda }\left(ZAZ^{†}B\right)\right\}=\hbar^{d}\frac{s_{\lambda }\left(A\right)s_{\lambda }\left(B\right)}{s_{\lambda }\left(p_{\infty }\right)},$$
*where $s_{\lambda }\left(p_{\infty }\right)$ is given in (Equation 8).*

*(II) Equivalently, for any $\Delta \in Y_{d}$ we have*
(32)$$E_{1,0}\left\{\;p_{\Delta }\left(ZAZ^{†}B\right)\;\right\}\;=\;z_{\Delta }\hbar^{d}\;\sum_{\Delta^{a},\Delta^{b}\in Y_{d}}\;H_{2}\left(\Delta ,\Delta^{a},\Delta^{b}\right)\;p_{\Delta^{a}}\left(A\right)\;p_{\Delta^{b}}\left(B\right),$$
*where*
(33)$$H_{2}\left(\Delta ,\Delta^{a},\Delta^{b}\right)=\sum_{\lambda \in Y_{d}}{\left(\frac{\dim\lambda }{d!}\right)}^{2}\varphi_{\lambda }\left(\Delta \right)\varphi_{\lambda }\left(\Delta^{a}\right)\varphi_{\lambda }\left(\Delta^{b}\right)$$
*is the three-point Hurwitz number (Equation 27).*


Note that Hurwitz numbers do not depend on the order of its arguments, in particular, for (Equation 33), $H_{2}\left(\Delta ,\Delta^{a},\Delta^{b}\right)=H_{2}\left(\Delta ,\Delta^{b},\Delta^{a}\right)=\cdots =H_{2}\left(\Delta^{b},\Delta^{a},\Delta \right)$.

**Proof.** (i) Relation (Equation 31) is known for Hermitian A,B; see [23], Chap. VII, Section 5, Example 5. Taking into account Remark 2, we see that both sides of (Equation 31) can be analytically continued as functions of matrix entries of *A* and *B*. Therefore, (Equation 31) is correct for $A,B\in {\mathrm{GL}}_{N}\left(C\right)$.(ii) Relation (Equation 32) follows from (Equation 31) and vice versa. To see this we use (Equation 10). For example, let us get (Equation 32) from (Equation 31). We replace all Schur functions in (Equation 31) with power sums in accordance with (Equation 7). Then we multiply the both sides of relation (Equation 31) by $\varphi_{\lambda }\left(\Delta \right)\dim\lambda$, then we sum over λ. Then the first orthogonality relation (Equation 10) and (Equation 27) results in (Equation 32).(iii) The alternative way to prove the Lemma is to start with the left-hand side of (Equation 32), where $A,B\in {\mathrm{GL}}_{N}\left(C\right)$. Such integrals were considered in [31] in the context of generating of Hurwitz numbers (see Appendix C). Using the results of [31] we state that it is equal to the right-hand side of (Equation 32), where $H_{2}\left(\Delta ,\Delta^{a},\Delta^{b}\right)$ is the three-point Hurwitz number given by (Equation 27), namely, to right-hand side of (Equation 33). Then with the help of orthogonality relation (Equation 10), we derive (Equation 31), where $A,B\in {\mathrm{GL}}_{N}\left(C\right)$. □

**Remark** **6.**
*Regarding the last point (iii): The volume of the article does not allow describing the construction of work [31,32]. In short: The derivation of the formula (Equation 33), and the derivation of more general formulas (Equation 68) below, are based on the application of Wick’s theorem and the realization of the fact that each Wick pairing corresponds to a certain gluing of surfaces from polygons: the surface obtained in the first order of the perturbation theory (|Δ|=d=1) is basic. (For instance, the torus can be obtained from a rectangular by gluing the opposite sides; see (a) and (b) in Figure 3.) The surfaces obtained in the following orders (d>1) are the surfaces which cover the basic one. In this case, the Hurwitz numbers simply give a weighted number of possible covering surfaces, where the Young diagrams correspond to the so-called branching profiles. The basic surface in case (Equation 33) is a sphere, glued from the 2-gon; see Figure 2b below where $X_{1}=Z,\;X_{-1}=Z^{†},\;C_{1}=A,\;C_{-1}=B$.*


Examples of (Equation 33). Suppose Δ=(2) on the left-hand side of (Equation 33). It is known that $H_{2}\left(\left(2\right),\Delta^{a},\Delta^{b}\right)=1/2$ in two cases—where $\Delta^{a}=\left(2\right),\;\Delta^{b}=\left(1,1\right)$ and where $\Delta^{a}=\left(1,1\right),\;\Delta^{b}=\left(2\right)$—and it is zero otherwise. Additionally $z_{\left(2\right)}=2$, see (Equation 1). Thus,
$$E_{1,0}\left\{\;\mathrm{tr}\left({\left(ZAZ^{†}B\right)}^{2}\right)\;\right\}\;=\;\hbar^{2}\;\mathrm{tr}\left(A^{2}\right)\;{\left(\mathrm{tr}B\right)}^{2}+\;\hbar^{2}\;{\left(\mathrm{tr}A\right)}^{2}\;\mathrm{tr}\left(B^{2}\right).$$

Next, suppose Δ=(1,1). As we know, $H_{2}\left(\left(1,1\right),\left(2\right),\left(2\right)\right)=1/2$ and vanishes for different choices of $\Delta^{a},\Delta^{b}$, and $z_{\left(1,1\right)}=2$; see (Equation 1); therefore,
$$E_{1,0}\left\{\;{\left(\mathrm{tr}ZAZ^{†}B\right)}^{2}\;\right\}\;=\;\hbar^{2}\;\mathrm{tr}\left(A^{2}\right)\;\mathrm{tr}\left(B^{2}\right).$$

**Corollary** **1.**
*Suppose detA≠0, detB≠0, and $\lambda =(\lambda_{1},\ldots ,\lambda_{N})$, ℓ(λ)≤N. We get*
(34)$$E_{1,0}\left\{s_{\lambda }\left(ZAZ^{†}B\right)\det\left({\left(Z^{†}Z\right)}^{p_{0}}\right)\right\}=\hbar^{d+p_{0}N}\frac{s_{\lambda }\left(A\right)s_{\lambda }\left(B\right)}{s_{\lambda }\left(p_{\infty }\right)}\frac{{\left(N+p_{0}\right)}_{\lambda }}{{\left(N\right)}_{\lambda }}$$

*In particular,*
(35)$$E_{1,0}\left\{s_{\lambda }\left(ZAZ^{†}\right)\det\left({\left(Z^{†}Z\right)}^{p_{0}}\right)\right\}=\hbar^{d+p_{0}N}s_{\lambda }\left(A\right){\left(N+p_{0}\right)}_{\lambda }$$


**Proof.** In case $p_{0}$ is a natural number the proof is as follows:
$$s_{\lambda }\left(ZAZ^{†}B\right)\det\left({\left(ZAZ^{†}B\right)}^{p_{0}}\right)=s_{\lambda }\left(ZAZ^{†}B\right)\det\left({\left(ZZ^{†}\right)}^{p_{0}}\right)\det\left({\left(A\right)}^{p_{0}}\right)\det\left({\left(B\right)}^{p_{0}}\right)=s_{\lambda +p_{0}}\left(ZAZ^{†}B\right)$$
and from Lemma 1: where $\lambda +p_{0}=\left(\lambda_{1}+p_{0},\lambda_{2}+p_{0},\ldots ,\lambda_{N}+p_{0}\right)$, $\ell (\lambda^{\prime })=N$. We have
(36)$$\frac{s_{\lambda }\left(p_{\infty }\right)}{s_{\lambda^{\prime }}\left(p_{\infty }\right)}=\prod_{i=1}^{N}\frac{\Gamma (h_{i}+1+p_{0})}{\Gamma (h_{i}+1)}=\frac{{\left(N+p_{0}\right)}_{\lambda }}{{\left(N\right)}_{\lambda }}$$As we see he right-hand side can by analytically continued as the function of $p_{0}$. □

**Lemma** **3.**
*(I) Suppose $\lambda \in Y_{d}$, and let ν be any partition. We have*
(37)$$E_{1,0}\left\{s_{\lambda }\left(ZA\right)s_{\nu }\left(Z^{†}B\right)\right\}=\hbar^{d}\delta_{\lambda \nu }\frac{s_{\lambda }\left(AB\right)}{s_{\lambda }\left(p_{\infty }\right)}.$$
*where $s_{\lambda }\left(p_{\infty }\right)$ is given in (Equation 8).*

*(II) Equivalently, let $\Delta^{a}\in Y_{d}$ and $\Delta^{b}\in Y_{d^{\prime }}$. Then*
(38)$$E_{1,0}\left\{\;p_{\Delta^{a}}\left(ZA\right)\;p_{\Delta^{b}}\left(Z^{†}B\right)\;\right\}\;=\;z_{\Delta^{a}}z_{\Delta^{b}}\;\delta_{d,d^{\prime }}\hbar^{d}\;\sum_{\Delta \in Y_{d}}\;H_{2}\left(\Delta^{a},\Delta^{b},\Delta \right)\;p_{\Delta }\left(AB\right),$$
*where $H_{{\mathrm{CP}}^{1}}\left(\Delta ,\Delta^{a},\Delta^{b}\right)$ is given by (Equation 33).*


The identity (Equation 37) is well-known; for instance, see [23]. Relation (Equation 38) was proven independently in [31]. To obtain (Equation 37) from (Equation 38) we replace power sums under the integral with the left-hand side of (Equation 6) and use (Equation 27) and the orthogonality relations (Equation 10) and (Equation 11).

Denote N×N identity matrix $I_{N}$. The following equality is known (see [23]: combine Examples 4 and 5 of Section 3 and Example 1 from Section I of Chapter I):(39)$$s_{\lambda }\left(I_{N}\right)={\left(N\right)}_{\lambda }s_{\lambda }\left(p_{\infty }\right).$$

Here
(40)$${\left(N\right)}_{\lambda }:={\left(N\right)}_{\lambda_{1}}{\left(N-1\right)}_{\lambda_{2}}\cdots {\left(N-\ell +1\right)}_{\lambda_{\ell }},$$
where $\lambda =(\lambda_{1},\ldots ,\lambda_{\ell })$, $\lambda_{\ell }\neq 0$ and where ${\left(a\right)}_{k}:=a\left(a+1\right)\cdots \left(a+k-1\right)=\frac{\Gamma (a+k)}{\Gamma (a)}$ is the Pochhammer symbol.

**Lemma** **4.**
*For any $\lambda \in Y_{d}$, we have*
(41)$$E_{0,1}\left\{s_{\lambda }\left(UAU^{-1}B\right)\right\}=\frac{s_{\lambda }\left(A\right)s_{\lambda }\left(B\right)}{s_{\lambda }\left(I_{N}\right)}\;.$$


The proof is contained in [23] Chap VII, Section 5, Example 3.

**Lemma** **5.**
*Suppose $\lambda \in Y_{d}$. For any μ, we get*
(42)$$E_{0,1}\left\{s_{\mu }\left(UA\right)s_{\lambda }\left(U^{-1}B\right)\right\}=\frac{s_{\lambda }\left(AB\right)}{s_{\lambda }\left(I_{N}\right)}\delta_{\mu ,\lambda }\;.$$


The proof follows from relations (1) and (3) Chap VII, Section 5, Example 1 in [23].

**Remark** **7.**
*Formula (Equation 41) gives the fastest derivation of the famous formula HCIZ:*
$$\int e^{\alpha \mathrm{tr}UAU^{†}B}d_{*}U=c\frac{\det{\left[e^{a_{i}b_{j}}\right]}_{i,j}}{\prod_{i>j}\left(a_{i}-a_{j}\right)\left(b_{i}-b_{j}\right)}$$

*Indeed, we use (Equation 19) where $p^{1}=\left(1,0,0,\ldots \right)=:p_{\infty }$ and $p^{2}=\left(p_{1}^{\left(2\right)},p_{2}^{\left(2\right)},\ldots \right)$ with $p_{m}^{\left(2\right)}=\mathrm{tr}\left({\left(UAU^{†}B\right)}^{m}\right)$; thus,*
$$e^{\alpha \mathrm{tr}UAU^{†}B}=\sum_{\lambda }\alpha^{\left|\lambda \right|}s_{\lambda }\left(p_{\infty }\right)s_{\lambda }\left(UAU^{†}B\right)$$
*which gives*
$$\int e^{\alpha \mathrm{tr}UAU^{†}B}d_{*}U=\sum_{\lambda }\alpha^{\left|\lambda \right|}\frac{s_{\lambda }\left(p_{\infty }\right)s_{\lambda }\left(A\right)s_{\lambda }\left(B\right)}{s_{\lambda }\left(I_{N}\right)}=\sum_{\lambda }\frac{\alpha^{\left|\lambda \right|}}{{\left(N\right)}_{\lambda }}s_{\lambda }\left(A\right)s_{\lambda }\left(B\right)$$
*which has the form (Equation 15). In a similar waym one evaluates $\int {\left(1-\alpha UAU^{†}B\right)}^{-a}d_{*}U$ and a number of integrals; see, for instance [26].*


The expectation values $E_{0,1}\left\{p_{\Delta }\left(AUBU^{-1}\right)\right\}$ and $E_{0,1}\left\{p_{\Delta^{a}}\left(AU\right)p_{\Delta^{b}}\left(BU†\right)\right\}$ can also be expressed in terms of Hurwitz numbers, but this requires a lot of space, and we will not do this; see the last section [31] for details.

## 3. Our Models. Products of Random Matrices We Choose

### 3.1. Preliminary. On the Products of Random Matrices

In a number of articles the spectral correlation functions of certain products of were were studied. These are products
(43)$$Z_{1}Z_{1}^{†}\cdots Z_{n}Z_{n}^{†},$$
(44)$$Z_{1}\cdots Z_{n}{\left(Z_{1}\cdots Z_{n}\right)}^{†},$$
or, a pair of products
(45)$$Z_{1}\cdots Z_{n},\;{\left(Z_{1}\cdots Z_{n}\right)}^{†};$$
see [10,11,12].

Similar products in which complex matrices are replaced by unitary and certain generalizations have also been considered [16,21,22,33,34,35].

Below we suggest a generalization of these models (see Examples 18 and 16 respectively for (Equation 43)–(Equation 45)). We call it matrix models related to dessin d’enfants, more precisely, to the so-called clean dessin d’enfants (see [36]).

This is a child’s drawing of a “constellation”, like the Greek constellation, which is painted in the sky with stars, and the sky is a surface with a chosen Euler characteristic.

See below for a more accurate description.

As a part of dessin d’enfants we will need the following modification:

(1) We shall consider mixed ensembles, and in what follows, $X_{\pm i}$ denotes either $Z_{\pm i}$, where $Z_{-i}=Z_{i}^{†}$, or $U_{\pm i}$, where $U_{-i}=U_{i}^{†}$.

(2) We need additional data, which we call source matrices $C_{\pm 1},\ldots ,C_{\pm n}\in {\mathrm{GL}}_{N}\left(C\right)$. Each random matrix $X_{i}$ enters only in combination $X_{i}C_{i}$:(46)$$X_{i}\;\to \;X_{i}C_{i}$$

Such combinations are very helpful (in particular, this makes it possible to consider rectangular random matrices). We shall use the notion of the *dressed* source matrices $C_{i},\;i=\pm 1,\ldots ,\pm n$ with the notation $L_{X}\left[*\right]$:(47)$$L_{X}\left[f\left(C_{1},C_{-1}\ldots ,C_{n},C_{-n}\right)\right]:=f\left(X_{1}C_{1},X_{-1}C_{-1}\ldots ,X_{n}C_{n},X_{-n}C_{-n}\right)$$
where *f* is any function. (The letter L is used to remember that it is the dressing from the left side.)

The matrices $\left\{C_{i}\right\}$ we call the source matrices which play the role of coupling constants in the matrix models below.

### 3.2. Models Obtained from Graphs (Geometrical View)

Consider a connected graph Γ on an orientable connected surface Σ without boundary with Euler characteristic e. We require such properties of the graph:

(1) its edges do not intersect. For example, the edges of the graph in Figure 1a do not intersect: the fact is that the graph is drawn on a torus, and not on a piece of paper.

(2) if we cut the surface Σ along the edges of the graph, then the surface will decompose into disks (more precisely: into pieces homeomorphic to disks) (see Figure 3a,b as an example).

As an example, see Figure 1, which contains all such graphs with two edges.

Such a graph is sometimes called a (clean) dessin d’enfants (the term dessin d’enfants without the additional “clean” serves for such a graph with a bipartite structure), sometimes—a map [28].

Let our graph have f
faces,
*n*
edges and V vertices; then
e = V − *n* + f.

We number all stars (i.e., all vertices of the graph Γ) with numbers from 1 to V and all faces of Γ with numbers from 1 to f
and all edges of Γ with numbers from 1 to
*n*
in any way.

We will slightly expand the vertices of the graph and turn them into the *small disk*, which sometimes for the sake of visual clarity we will call *stars*.

In this case, the edges coming out of the vertex will divide the border of a small disk into segments. We orient the boundary of each such segment clockwise, and the segment can be represented by an arrow that goes from one edge to another; see the Figure 2a,b,d and Figure 3a,c as examples. Our graph will have a total of 2n arrow segments.

It is more correct to assume that the edges of the graph are very thin ribbons; that is, they have finite thickness. That is, the edge number |i| has two sides, one of which we number with the number *i*, and the other with the number −i (this choice is arbitrary but fixed). It would be more correct to depict each edge of Γ numbered by |i| in the form of a ribbon, the sides of which are two oppositely directed arrows; one arrow has a number *i*, and the second −i. However, with the exception of Figure 3, we did not do that so as not to clutter up the drawings.

Now, each arrow of the side of the edge rests with its end against the arrow-segment—as if continued by the arrow-segment. We assign the same number *i* (i=±1,…,±n) to the side of the edge Γ and to the segment of the small disk, which continues this side while traversing the face in the positive direction.

Additionally, to the number *i* (i=±1,…,±n) we attribute the product of the N×N matrices
$$i\;\to \;X_{i}C_{i}$$
where $C_{i}$ is assigned to the *i* segment of a small disk and will be called *source matrix*, and $X_{i}$ is assigned to the arrow *i*, which is the side of the edge |i| of the graph Γ; see Figure 2 for an illustration.

Let us use the following numbering. If one side of the edge of the tape |i| is labeled *i*, then the other side of the same edges is labeled −i. (It does not matter which side of the edge |i| we assign the number *i* to, and which we assign the number −i to, but we should fix the numbering we have chosen). The two sides of an edge of the graph Γ are actually arrows pointing in opposite directions. (In order not to complicate the drawing, we do not depict these arrows in our figures, except for Figure 3a,b). We draw the arrows so that they point in the positive direction when bypassing the boundary of the face (that is, when bypassing the face in the counterclockwise direction). Each arrow, say *i* (which either a positive or a negative number), ends at the boundary of the small disk (star) and with further bypassing of the face we pass along the segment of the boundary of the star. We assign the same number *i* to this segment. Acting in this way, we give numbers to all segments of all small disks. It is easy to understand that all the arrows on the segments of the boundary of any small disc are directed clockwise (i.e., in the negative direction) when passing along the boundary of each small disk.

We attribute a sequential set of numbers to each star as follows. (this set is defined up to a cyclic permutation, and we will call it a cycle associated with the star). Examples: These are numbers (1,−1) assigned to the star (yellow small disk) in Figure 2a. These are the numbers (k,−j) that we attribute to the star on the top in the Figure 2c and the numbers (−k,j,−i) that we attribute to the lower star in the same figure. Additionally, to each cycle we ascribe the related cyclic products:$$\left(1,-1\right)\;↔\;C_{1}C_{-1},\;$$
to the star (yellow small disk) in Figure 2a. As for the stars in Figure 2c we obtain
$$\left(k,-j\right)\;↔\;C_{k}C_{-j}\;\;\left(-k,j,-i\right)\;↔\;C_{-k}C_{j}C_{-i}$$

Each cycle product we will call the *star’s monodromy*. As a result, a star’s monodromy is a product of matrices that are attributed to arrow segments, taken in the same sequence in which arrows follow each other when moving around a small disk clockwise. We number the stars in any way by numbers from 1 to V. The monodromy of a star *i* will be denoted by the letter $W_{i}^{*}$.

In addition to the edges and in addition to the vertices, we number the faces of the graph Γ and the corresponding face monodromies with numbers from 1 to f. The cycles corresponding to the face
*i*
will be denoted by 
*f_i_*
and defined as follows. When going around the face border in the positivedirection, namely, counterclockwise for an ordinary face (or counterclockwise if the face contains infinity), wecollect segment numbers of small disks;for a face with a number
*i*, this ordered collection of ordered numbers is
*f_i_*.As in the case of stars, we build cyclic products
*f_i_* ↔ *W_i_*. Examples:
$$f_{1}=\left(1\right)\;↔\;C_{1}=W_{1}\;\mathrm{and}\;f_{2}=\left(-1\right)\;↔\;C_{-1}=W_{2}$$
for two faces in
Figure 2a.

We also introduce *dressed* monodromies:$$L_{X}\left(W_{1}\right)=X_{1}C_{1},\;L_{X}\left(W_{-1}\right)=X_{-1}C_{-1}$$

Additionally, for the face in Figure 2c:$$f_{1}=\left(-j,-k\right)\;↔\;C_{-j}C_{-k}=W_{1}\;↔L_{X}\left(W_{1}\right)=X_{-j}C_{-j}X_{-k}C_{-k}$$

Thus, we have two sets of cycles and two sets of monodromies: vertex cycles and vertex monodromies
(48)$$\sigma_{i}^{-1}\;↔\;W_{i}^{*},\;i=1,\ldots ,V$$
and face cycles and face monodromies: The cycles corresponding to the face *i* will be denoted by $f_{i}$; we have
(49)$$f_{i}\;↔\;W_{i}\;↔\;L_{X}\left[W_{i}\right],\;i=1,\ldots ,F$$

Let us note that both cycles and monodromies are defined up to the cyclic permutation.

**Remark** **8.**
*Important remark. Please note that each of the matrices $C_{i},\;i=\pm 1,\ldots ,\pm n$ enters the set of monodromies $W_{1},\ldots ,W_{f}$ once and only once. Accordingly, each random matrix $X_{i},\;i=\pm 1,\ldots ,\pm n$ is included once and only once in the set of dressed monodromies $L_{X}\left[W_{1}\right],\ldots ,L_{X}\left[W_{f}\right]$. This determines the class of matrix models that we will consider and which we will call matrix models of dessin d’enfants.*


**Remark** **9.**
*In the description of maps (the same, of clean dessin d’enfants) the following combinatorial relation is well-known; see the wonderful texbook [28], Remark 1.3.19:*
(50)$$\left(\prod_{i=1}^{n}\alpha_{i}\right)\left(\prod_{i=1}^{F}f_{i}\right)=\left(\prod_{i=1}^{V}\sigma_{i}^{-1}\right)$$
*where each of $\alpha_{i}$, $f_{i}$ and $\sigma_{i}$ belongs to the permutation group $S_{2n}$. Here $f_{i}$ are face cycles, $\sigma_{i}$ are vertex cycles and $\left(\prod_{i=1}^{n}\alpha_{i}\right)$ is an involution without fixed points; each $\alpha_{i}$ transposes i and −i. Since $\alpha_{i}^{2}=1$ we can also write*
(51)$$\left(\prod_{i=1}^{n}\alpha_{i}\right)\left(\prod_{i=1}^{V}\sigma_{i}^{-1}\right)=\left(\prod_{i=1}^{F}f_{i}\right)$$

*The reader can check this relation for the pair of the simplest examples where n=1.*

*For any given set of the face cycles $f_{1},\ldots ,f_{F}$, the relation (50) allows you to uniquely reconstruct the set of the vertex cycles $\sigma_{1}^{-1},\ldots ,\sigma_{V}^{-1}$ (the power −1 is related to the negative (clockwise) counting of numbers).*


We get

**Proposition** **1.**(52)$$\hbar^{-n_{1}}E_{n_{1},n_{2}}\left\{\prod_{i=1}^{F}L_{X}\left[\mathrm{tr}W_{i}\right]\right\}=N^{-n_{2}}\prod_{i=1}^{V}\mathrm{tr}W_{i}^{*}$$*and*(53)$$\hbar^{-n_{1}}E_{n_{1},n_{2}}\left\{\prod_{i=1}^{V}L_{X}\left[\mathrm{tr}W_{i}^{*}\right]\right\}=N^{-n_{2}}\prod_{i=1}^{F}\mathrm{tr}W_{i}\;,$$*where we use the notation (47) and where the parameters**n*_1_, *n*_2_, *ℏ were defined in Section 2.2.*

We note that each trace is a sum of monomials. Let us note that each of $X_{i},\;i=\pm 1,\ldots ,\pm n$ enters only once in each monomials term insider of the integral in the left and side of (Equation 52) and of (Equation 53).

There are two ways to prove these relations: algebraic and geometrical ones. In short, the sketches of the proof are are as follows. We should notice that $p_{\left(1\right)}\left(*\right)=s_{\left(1\right)}\left(*\right)$ and apply Lemmas 2–5 while having in mind that this is actually the cut-or-join procedure (Equation 57) stated below in Section 3.3. In this case the difference between Lemmas 2, 3 and Lemmas 4, 5 is only in the power of the factor *N*, since ${\left(N\right)}_{\lambda }=N$ in case λ=(1).

Geometric way: We use the relation
$$\hbar^{-1}E_{n_{1},0}\left\{{\left(Z_{i}\right)}_{a,b}{\left(Z_{j}^{†}\right)}_{b^{\prime },a^{\prime }}\right\}=\delta_{i,j}\delta_{a,a^{\prime }}\delta_{b,b^{\prime }}$$
and calculate monodromies along faces of the graph Γ as described in the beginning of this section.

**Remark** **10.**
*From geometrical point of view in this way we create a surface by gluing the polygons related to the dressed face monodromies; see [31].*


**Remark** **11.**
*You may notice some similarities between formulas (Equation 52), (Equation 53) and formulas (Equation 50), (Equation 51). Additionally, there is. One can say that the role of involution $\alpha_{i}$ is played by the integration over the matrix $X_{i}$ (by the Wick coupling of $X_{i}$ and $X_{-i}$).*


We draw attention to two facts.

The answer (i.e., the left-hand side of (Equation 52)) depends only on the spectrum of star monodromies
$$\mathrm{Spect}\left(W_{1}^{*}\right),\ldots ,\mathrm{Spect}\left(W_{V}^{*}\right).$$The answer does not depend on how exactly in our model we distribute the matrices {Z} and {U} to dress the source matrices—only two numbers $n_{1}$ and $n_{2}$ are important. For example, it does not matter, in the right-hand side of (Equation 52), whether we dress $L_{X}\left[C_{i},C_{-i}\right]=U_{i}C_{i},U_{-i}C_{-i}$ and $L_{X}\left[C_{j},C_{-j}\right]=Z_{j}C_{j},Z_{-j}C_{-j}$ or $L_{X}\left[C_{i},C_{-i}\right]=Z_{i}C_{i},Z_{-i}C_{-i}$ and $L_{X}\left[C_{j},C_{-j}\right]=U_{j}C_{j},U_{-j}C_{-j}$.

### 3.3. Our Models. Algebraic View

You can forget about graphs and dessin d’enfants and set them out differently.

We have a set of random matrices
$$X_{\pm 1},\ldots ,X_{\pm n},\;X_{-i}=X_{i}^{†}$$
and there are as many source matrices
(54)$$C_{\pm 1},\ldots ,C_{\pm n},$$
and we want to consider expectation value of the products of these matrices. We want each matrix $X_{i}$ to come in combination with the source matrix labeled with the same *i*; see (Equation 46) and (Equation 47).

The set (Equation 54) we will call the alphabet of pairs and the matrices $\left\{C_{i}\right\}$ can be considered letters of the alphabet.

The products of these matrices can be considered as words constructed from letters of the alphabet of pairs. Consider a group of words
$$W_{1},\ldots ,W_{F}$$
with the following condition: each of matrices $C_{i},\;i=\pm 1,\ldots ,\pm n$ is included *once and only once* in one of the words of this group.

In addition, we ask the following property: In this group of words there is no such subset of words that could be constructed from the alphabet of pairs with a smaller set of pairs of letters (in other words, we will consider only connected groups of words. The connected group of words will be related to the connected graphs Γ).

Each word $W_{i}$ can be associated with an ordered set of numbers $f_{i}$ consisting of the numbers of the matrices included in the word: $f_{i}↔W_{i}$. Recall that each word $W_{i}$ (and respectively $f_{i}$) is defined up to a cyclic permutation.

There is a one-to-one correspondence between dessin d’enfants with *n* edges and f faces and word sets constructed in this way. Based on the set $W_{1},\ldots ,W_{F}$, the surface Σ is built uniquely. To do this, the procedure is as follows. First, for each word, say $W_{i}$, we associate the polygon with the edges, numbered by the numbers of the matrices in the product, namely, by the set $f_{i}$. By going around the boundary of the polygon counterclockwise, we assign the numbers of the matrices to the edges if we write the word from left to right. We get a set of f polygons. After that, we glue these polygons so that the side with the number i sticks together with the side with the number −i. We assign an orientation to each polygon, considering its edges arrows, showing the direction of counterclockwise. We glue the edges so that the beginning of the arrow i sticks together with the end of the arrow −i. We get the surface Σ and on it is the graph Γ, whose edges are glued from two oppositely directed arrows (ribbon edge).

To determine the Euler characteristic of Σ we need to know the number of vertices of the graph Γ.

Cut-or-join procedure. Purely algebraically, one should act like this.

Consider $W_{1}⊗W_{2}⊗\cdots ⊗W_{F}$ (the order in this tensor product is not important) and the set of involutions $T_{i},i=1,\ldots ,n$, which act on this tensor product as follows. Each involution of $T_{i}$ does not affect those $W_{a}$ that contain neither $C_{i}$ nor $C_{-i}$. Two situations are possible. (I) The matrices $C_{i}$ and $C_{-i}$ are in the same word, say, the word $W_{a}$. How then can we to rearrange the matrices with the word cyclically, to bring it to the form $C_{i}XC_{-i}Y$, where X and Y are some matrices. (II) The matrices $C_{i}$ and $C_{-i}$ are included in different words; in this case we will write these two words as $C_{i}X$ and $C_{-i}Y$. As you can see, the action of involutions $T_{i}$ corresponds to taking the integral in Lemmas 2–5. Then
(55)$$T_{i}\left[\cdots ⊗C_{i}XC_{-i}Y⊗\cdots \right]=\cdots ⊗C_{i}X⊗C_{i}Y⊗\cdots$$
(56)$$T_{i}\left[\cdots ⊗C_{i}X⊗C_{i}Y⊗\cdots \right]=\cdots ⊗C_{i}XC_{-i}Y⊗\cdots$$

It is easy to see that involutions commute: $T_{i}\left[T_{j}\left[*\right]\right]=T_{j}\left[T_{i}\left[*\right]\right]$.

The transformation
(57)$$W_{1},\ldots ,W_{F}↔\;W_{1}^{*},\ldots ,W_{V}^{*}$$
can be obtained purely algebraically in *n* steps. We call these two sets of matrices *dual sets*.

Proposition:
(58)$$\begin{array}{c} \prod_{i=1}^{n}T_{i}\left[W_{1}⊗W_{2}⊗\cdots ⊗W_{F}]=W_{1}^{*}⊗\cdots ⊗W_{V}^{*}\right. \end{array}$$
(59)$$\begin{array}{c} \prod_{i=1}^{n}T_{i}\left[W_{1}^{*}⊗\cdots ⊗W_{V}^{*}\right]=W_{1}⊗W_{2}⊗\cdots ⊗W_{F} \end{array}$$

**Remark** **12.**
*Actually this is a manifestation of the Equation (Equation 50) where $\alpha_{i}$ is related to $T_{i}$. An involution without fixed points $\prod_{i=1}^{n}T_{i}$ takes the graph *Γ* to the graph dual to it.*


Here are some examples (Equation 57) with explanations about the geometric representation in Figure 1, Figure 2, Figure 3, Figure 4 and Figure 5 and in some other graphs:
**Example** **1.***Suppose $W_{1}=C_{1}$, $W_{2}=C_{-1}$. In 1 step we get:*$$W_{1}⊗W_{2}=C_{1}⊗C_{-1}↔C_{1}C_{-1}=W_{1}^{*}$$Therefore V=1 and as we have F=2,n=1; then the Euler characteristic is E=F−n+V=2. The right hand side can be obtained from Figure 2a as the face monodromies while the left-hand side can be obtained as star monodromies there. The left-hand side contains also the face monodromies in Figure 2b; and the right-hand side can be obtained as star monodromies there. (This is because graphs (a) and (b) are dual ones.)
**Example** **2.***In 2 steps:*$$W_{1}=C_{1}C_{2}C_{-1}C_{-2}↔C_{1}C_{-2}C_{-1}C_{2}=W_{1}^{*}$$Therefore V=1, and as we have F=1,n=2, then the Euler characteristic is E=F−n+V=0 (torus). The right-hand side can be obtained from Figure 1a as the face monodromies while the left-hand side can be obtained as star monodromies there; see the more detailed Figure 3a. The dual graph looks like the same as the graph on the torus: there is 1 vertex and 1 face; the left-hand side plays the role of the face monodromy for the dual graph. This a particular case of Ex 9.
**Example** **3.***In two steps:*$$W_{1}⊗W_{2}⊗W_{3}=C_{-1}C_{-2}⊗C_{1}⊗C_{2}↔C_{1}C_{-1}C_{2}C_{-2}=W_{1}^{*}$$Therefore V=1 and as we have F=3,n=2, the Euler characteristic is E=F−n+V=2. The right-hand side can be obtained from Figure 1b as the face monodromies while the left-hand side can be obtained as star monodromies there. The left-hand side contains also the face monodromies in Figure 1c (see Figure 3c for more details); the right-hand side can be obtained as star monodromies there. (This is because Figure 1b and Figure 1c are dual ones).
**Example** **4.***In 2 steps:*$$C_{1}C_{2}C_{-1}⊗C_{-2}↔C_{1}C_{-2}C_{2}⊗C_{-1}$$Therefore V=2 and as we have F=2,n=2, the Euler characteristic is E=V−n+V=2. The right-hand side can be obtained from Figure 1e as the face monodromies while the left-hand side can be obtained as star monodromies there. The left-hand side contains also the face monodromies of the dual graph which is the graph of the same type (the loop from which the segment sticks out).
**Example** **5.***In 5 steps:*$$W_{1}⊗W_{2}⊗W_{3}=C_{1}C_{2}C_{3}C_{4}⊗C_{-3}C_{-2}C_{5}⊗C_{-5}C_{-1}C_{-4}↔C_{2}C_{-2}C_{-5}⊗C_{2}C_{-3}⊗C_{5}C_{3}C_{-4}⊗C_{4}C_{-1}=W_{1}^{*}⊗W_{2}^{*}⊗W_{3}^{*}⊗W_{4}^{*}$$Therefore V=4 and as we have F=3,n=5, the Euler characteristic is E=F−n+V=2 (the same is obtained from Figure 5d).
**Example** **6.***In n steps:*$$C_{1}C_{2}\cdots C_{n}C_{-n}\cdots C_{-1}↔C_{1}C_{-2}⊗C_{2}C_{-3}⊗\cdots ⊗C_{n-1}C_{-n}⊗C_{n}⊗C_{-1}$$Therefore V=n+1 and as we have *F = 1*, the Euler characteristic is E=F−n+V=2 The left-hand side is related to *Γ* in form of a chain with n+1 vertices, n edges drawn on the sphere (see Figure 1b for the n=2 chain and see Figure 4a for the n=3 chain). In case each source matrix is the identity ones, this is related to (Equation 44) which is a rather popular product. The right side is represented by such a graph: there are n circles; each subsequent one decreases and is inside the previous one. They all touch at one point (the vertex); see Figure 4b.
**Example** **7.***In n steps:*$$C_{1}C_{2}\cdots C_{n}⊗C_{-n}C_{1-n}\cdots C_{-1}↔C_{1}C_{-2}⊗C_{2}C_{-3}⊗\cdots ⊗C_{n-1}C_{-n}⊗C_{n}C_{-1}$$Therefore V=n, and as we have *f = 2*, then the Euler characteristic is E=F−n+V=2. (the case n=2 is obtained from Figure 1d). The left-hand side is related to *Γ* in form of the chain from the previous case where we replace n by n−1 and connect its ends by n-th edge. Namely, it is n-gon drawn on the sphere. The right-hand side is related to the graph where two vertices are connected by n edges. In case n=4, the right-hand side can be obtained from Figure 4d as the face monodromies while the left-hand side can be obtained as star monodromies there. The left-hand side contains also the face monodromies in Figure 4c; the right-hand side can be obtained as star monodromies there. (Indeed, graphs in Figure 4d and in Figure 4c are dual ones.)
**Example** **8.***In n steps:*$$C_{1}C_{-1}C_{2}C_{-2}\cdots C_{n}C_{-n}↔C_{-1}C_{-2}\cdots C_{-n}⊗C_{1}⊗C_{2}⊗\cdots ⊗C_{n}$$Therefore V=n+1, and as we have *F = 1*, then the Euler characteristic is E=F−n+V=2. The left-hand side is related to a star-graph *Γ*; the right-hand side—to a petal graph (as an example take n=3 and look at Figure 5a and dual Figure 5b.After the dressing procedure, in case all source matrices are chosen to be identical ones we get a popular product (Equation 43).
**Example** **9.***Suppose $W_{1}=C_{a_{1}}C_{b_{1}}C_{-a_{1}}C_{-b_{1}}\cdots C_{a_{g}}C_{b_{g}}C_{-a_{g}}C_{-b_{g}}$, where we number source matrices according $a_{i},b_{i}$-cycles structure of a Riemann surface of genus g. In n=2g steps we obtain*$$C_{a_{1}}C_{b_{1}}C_{-a_{1}}C_{-b_{1}}\cdots C_{a_{g}}C_{b_{g}}C_{-a_{g}}C_{-b_{g}}↔C_{-a_{1}}C_{b_{1}}C_{a_{1}}C_{-b_{1}}\cdots C_{-a_{g}}C_{b_{g}}C_{a_{g}}C_{-b_{g}}$$Therefore V=1, and as we have *F = 1*, n=2g, then the Euler characteristic is E=F−n+V=2−2g.The case n=2 yields a torus and was considered in Example 2. The case n=4 can be related to a graph drawn on a pretzel; see Figure 5e. The left-hand side and the right-hand sides are related to dual graphs each of which has one face and one vertex and is drawn on a surface with genus g whose edges are $a_{i},b_{i}$ cycles.
**Example** **10.***In n=6 steps we obtain*$$C_{-1}C_{-2}C_{-3}⊗C_{-5}C_{-4}C_{1}⊗C_{-6}C_{5}C_{2}⊗C_{6}C_{4}C_{3}↔C_{1}C_{4}C_{-3}⊗C_{-5}C_{-1}C_{2}⊗C_{-6}C_{-2}C_{3}⊗C_{-4}C_{5}C_{6}$$Therefore V=4, and as we have F=4,n=6, then the Euler characteristic is E=F−n+V=2. This can be represented as a tetrahedron inscribed in a sphere. The tetrahedron graph is self-dual.

## 4. Expectation Values of Matrix Products

Lemmas 2–5 are generalized to the case of mixed ensembles; see Propositions 2 and 4 below.

Propositions 2–4 are extended versions of statements studied in [31,32,37].

**Proposition** **2.**
*Consider ensemble (Equation 29). Consider dual sets $W_{1},\ldots ,W_{F}$ and $W_{1}^{*},\ldots ,W_{V}^{*}$ of (57). For any given set of partitions $\lambda^{1},\lambda^{2},\ldots ,\lambda^{F}=\lambda \in Y_{d}$, we have*
(60)$$E_{n_{1},n_{2}}\left\{L_{X}\left[s_{\lambda^{1}}\left(W_{1}\right)\cdots s_{\lambda^{F}}\left(W_{F}\right)\right]\right\}=\delta_{\lambda^{1},\lambda^{2},\ldots ,\lambda^{F}}\hbar^{n_{1}d}{\left(s_{\lambda }\left(p_{\infty }\right)\right)}^{-n_{1}}{\left(s_{\lambda }\left(I_{N}\right)\right)}^{-n_{2}}s_{\lambda }\left(W_{1}^{*}\right)\cdots s_{\lambda }\left(W_{V}^{*}\right),$$
*where $\delta_{\lambda^{1},\lambda^{2},\ldots ,\lambda^{F}}$ is equal to 1 in case $\lambda^{1}=\lambda^{2}=\cdots =\lambda^{F}$ and to 0 otherwise.*

*Similarly, for any set of partitions $\lambda^{1},\lambda^{2},\ldots ,\lambda^{V}=\lambda \in Y_{d}$, we get*
(61)$$E_{n_{1},n_{2}}\left\{L_{X}\left[s_{\lambda^{1}}\left(W_{1}^{*}\right)\cdots s_{\lambda^{F}}\left(W_{V}^{*}\right)\right]\right\}=\delta_{\lambda^{1},\lambda^{2},\cdots ,\lambda^{V}}\hbar^{n_{1}d}{\left(s_{\lambda }\left(p_{\infty }\right)\right)}^{-n_{1}}{\left(s_{\lambda }\left(I_{N}\right)\right)}^{-n_{2}}s_{\lambda }\left(W_{1}\right)\cdots s_{\lambda }\left(W_{F}\right),$$
*where $\delta_{\lambda^{1},\lambda^{2},\cdots ,\lambda^{F}}$ is equal to 1 in case $\lambda^{1}=\lambda^{2}=\cdots =\lambda^{F}$ and to 0 otherwise.*


The sketch of proof. The proof is based on the cut-or-join procedure (57) of Section 3.3. which is the result of the step-by-step application of Lemmas 2–5. The different (geometrical) proof is based on the treating of the Wick rule as a way to glue surfaces from polygons and the use of (11) and of (27), (28).

**Corollary** **2.**
*Consider ensemble (29). Consider dual sets $W_{1},\cdots ,W_{F}$ and $W_{1}^{*},\ldots ,W_{V}^{*}$ of (57), such that $\det W_{i},\det W_{j}^{*}\neq 0$ for i *= 1, …, F*, j *= 1, …, V*. Suppose that $\alpha_{m}=\max\left(\alpha^{1},\ldots ,\alpha^{F}\right)=:\alpha ,\;where\;\alpha^{1},\ldots ,\alpha^{F}$ is a given set of nonnegative integers. Consider a given set of partitions $\lambda^{i},\;i=1,\ldots ,F\;and\;denote\;\lambda =\lambda^{\left(m\right)}$. We have*
(62)$$E_{n_{1},n_{2}}\left\{L_{X}\left[\prod_{i=1}^{F}s_{\lambda^{i}}\left(W_{i}\right)\det\left({\left(W_{i}\right)}^{\alpha_{i}}\right)\right]\right\}=$$
(63)$$\delta_{\lambda^{1}+\alpha_{1},\lambda^{2}+\alpha_{2},\ldots ,\lambda^{F}+\alpha_{f}}{\left(\hbar^{-(|\lambda |+\alpha N)}s_{\lambda }\left(p_{\infty }\right)\frac{{\left(N\right)}_{\lambda }}{{\left(N+\alpha \right)}_{\lambda }}\right)}^{-n_{1}}{\left(s_{\lambda }\left(I_{N}\right)\right)}^{-n_{2}}\prod_{i=1}^{V}s_{\lambda }\left(W_{i}^{*}\right)\det\left({\left(W_{i}^{*}\right)}^{\alpha }\right)$$
*where $\delta_{\lambda^{1}+\alpha_{1},\lambda^{2}+\alpha_{2},\ldots ,\lambda^{F}+\alpha_{F}}$ is equal to 1 in case for each k *= 1*, …, N we have $\lambda_{k}^{\left(1\right)}+\alpha_{1}=\lambda_{k}^{\left(2\right)}+\alpha_{2}=\cdots =\lambda_{k}^{\left(F\right)}+\alpha_{F}$, and it is 0 otherwise.*


**Proposition** **3**
**([32]).**
*let $\Delta^{i}=\left(\Delta_{1}^{i},\Delta_{2}^{i},\ldots \right),\;i=1,\ldots ,k$ be a set of partitions of the weights $d_{1},\ldots ,d_{k}=d.$*

*Let $\mu^{k+1}=(\mu_{1}^{\left(k+1\right)},\mu_{2}^{\left(k+1\right)},\ldots ),\ldots ,\mu^{\left(F\right)}=\mu =(\mu_{1},\mu_{2},\ldots )$, where *1 <* k *< F*, be a set of partitions. We get*
(64)$$E_{n_{1},n_{2}}\left\{L_{X}\left[\prod_{i=1}^{k}p_{\Delta^{i}}\left(W_{i}\right)\prod_{i=k+1}^{F}s_{\mu^{i}}\left(W_{i}\right)\right]\right\}=$$
(65)$$\delta_{|\mu |,d}\delta_{d_{1},\ldots ,d_{k}}\delta_{\mu^{k+1},\ldots ,\mu^{F-k}}\hbar^{n_{1}d}{\left({\left(N\right)}_{\mu }\right)}^{-n_{2}}{\left(\frac{\dim\;\mu }{d!}\right)}^{-n_{1}-n_{2}}\chi_{\mu }\left(\Delta^{1}\right)\cdots \chi_{\mu }\left(\Delta^{k}\right)\prod_{i=1}^{V}s_{\lambda }\left(W_{i}^{*}\right)$$
*where ${\left(N\right)}_{\mu }$ is given by (40) and $\chi_{\mu }\left(\Delta \right)$ is the character of the symmetric group; see Remark (5). The symbol $\delta_{\mu^{k+1},\ldots ,\mu^{F}}$ is equal to 1 in case $\mu^{k+1}=\cdots =\mu^{F-k}$ and is equal to 0 otherwise.*

*Similarly, let*
$\mu^{k+1},\ldots ,\mu^{V}=\mu ,\;where\;1\;<\;k\;<\;V,\;be\;a\;set\;of\;partitions.\;Then$
(66)$$E_{n_{1},n_{2}}\left\{L_{X}\left[\prod_{i=1}^{k}p_{\Delta^{i}}\left(W_{i}^{*}\right)\prod_{i=k+1}^{V}s_{\mu^{i}}\left(W_{i}^{*}\right)\right]\right\}=$$
(67)$$\begin{array}{c} \delta_{|\mu |,d}\delta_{d_{1},\ldots ,d_{k}}\delta_{\mu^{k+1},\ldots ,\mu^{V-k}}\hbar^{n_{1}d}{\left({\left(N\right)}_{\mu }\right)}^{-n_{2}}{\left(\frac{\dim\;\mu }{d!}\right)}^{-n_{1}-n_{2}}\chi_{\mu }\left(\Delta^{1}\right)\cdots \chi_{\mu }\left(\Delta^{k}\right)\prod_{i=1}^{F}s_{\lambda }\left(W_{i}\right) \end{array}$$


The Proposition is derived from the previous one using (5) and (Equation 11).

**Proposition** **4.**
*Let*
$\Delta^{i}=\left(\Delta_{1}^{i},\Delta_{2}^{i},\ldots \right),\;i=1,\ldots ,F\;be\;a\;set\;of\;partitions\;of\;weights\;d_{1},\ldots ,d_{F}=d.$

*Then*
(68)$$E_{n_{1},n_{2}}\left\{L_{X}\left[p_{\Delta^{1}}\left(W_{1}\right)\cdots p_{\Delta^{F}}\left(W_{F}\right)\right]\right\}\prod_{i=1}^{F}\frac{1}{z_{\Delta^{i}}}\;=$$
(69)$$\delta_{d_{1},\ldots ,d_{F}}\hbar^{n_{1}d}\sum_{{\tilde{\Delta }}^{1},\ldots ,{\tilde{\Delta }}^{V}\in Y_{d}}p_{{\tilde{\Delta }}^{1}}\left(W_{1}^{*}\right)\cdots p_{{\tilde{\Delta }}^{V}}\left(W_{V}^{*}\right)H_{E}\left({\tilde{\Delta }}^{1},\ldots ,{\tilde{\Delta }}^{V},\Delta^{1},\ldots ,\Delta^{F}|n_{2}\right),$$
*where $\delta_{d_{1},\ldots ,d_{F}}=1\;in\;case\;d_{1}=\cdots =d_{F}\;and\;0\;otherwise,\;and\;Y_{d}\;is\;the\;set\;of\;all\;partitions\;of\;weight\;d.\;The\;prefactor\;H_{E}\left({\tilde{\Delta }}^{1},\ldots ,{\tilde{\Delta }}^{V},\Delta^{1},\ldots ,\Delta^{F}|n_{2}\right)$ in (69) is the weighted Hurwitz number (28) with k *= F + V* and *E = F −* n *+ V*.*

*Similarly, for a given set of partitions*
${\tilde{\Delta }}^{i}=({\tilde{\Delta }}_{1}^{i},{\tilde{\Delta }}_{2}^{i},\ldots ),\;i=1,\ldots ,V\;with\;weights\;d_{1},\ldots ,d_{V}=d\;respectively,\;we\;get$
(70)$$E_{n_{1},n_{2}}\{L_{X}[p_{{\tilde{\Delta }}^{1}}\left(W_{1}^{*}\right)\cdots p_{{\tilde{\Delta }}^{V}}\left(W_{V}^{*}\right)]\}=$$
(71)$$\delta_{d_{1},\ldots ,d_{V}}\hbar^{n_{1}d}\sum_{{\tilde{\Delta }}^{1},\ldots ,{\tilde{\Delta }}^{F}\in Y_{d}}p_{\Delta^{1}}\left(W_{1}\right)\cdots p_{\Delta^{V}}\left(W_{V}\right)H_{E}\left({\tilde{\Delta }}^{1},\ldots ,{\tilde{\Delta }}^{V},\Delta^{1},\ldots ,\Delta^{F}|n_{2}\right),$$
*where $H_{E}({\tilde{\Delta }}^{1},\ldots ,{\tilde{\Delta }}^{V},\Delta^{1},\ldots ,\Delta^{F})$ is exactly the same as in (69).*


Propositions 2–4 and are equivalent. This can be proven with the help of (10), (11) and (28).

**Remark** **13.**
*Proposition 4 was proved in [31] using a geometrical construction of Hurwitz numbers as a number of ways to glue polygons. Each matrix entry, say ${\left(X_{i}\right)}_{a,b}$ may be drawn as an arrow with labels a and b at the startpoint and the endpoint respectively. We draw solid arrow for an entry of a random matrix and a dashed arrow for an entry of a source matrix. The product of matrices we draw as arrows sequentially assigned to each other. The trace of a product is drawn as a polygon. Now each $L_{X}\left[trW_{c}\right]$ is a polygon with alternating solid and dashed-edge arrows; we orient the edges counterclockwise. In [31] we named such polygons countries. Thus, we relate each dressed word to a country. It may be shown that the expectation*
$$E_{n,0}\left\{L_{X}\left[tr\left(W_{1}\right)\cdots tr\left(W_{1}\right)\right]\right\}$$
*may be viewed as the result of gluing of the net of countries into a surface, say $\Sigma ,\;and\;E=F-n+V\;being\;the\;Euler\;characteristic\;of\;this\;surface.\;This\;is\;a\;result\;of\;the\;Gaussian\;integration\;over\;matrices\;Z_{i}.\;Oppositely\;directed\;solid\;arrows\;\left(corresponding\;to\;Z_{i}\;and\;Z_{i}^{†}\right)form\;sides\;of\;ribbon\;edges.\;These\;edges\;end\;at\;a\;boundary\;of\;disks\;\left(inflated\;vertices\right)\;with\;dashed\;boundaries\;\left(source\;matrices\;are\;attached\;to\;the\;segments\;of\;dashed\;boundaries\right).$ In [Equation 31] we named this disk watchtowers. There are n ribbon edges (the boarders of countries) and *2*n dashed edges (segments of boundaries of disks—of the boarders of the watchtowers); there are *V* watchtowers; and there are *F* countries with alternating (solid-dashed) edges. There are *2*n 3-valent vertices (ends of ribbons): there are two dashed arrows (one is outgoing; another is incoming) and one ribbon (one side is the solid outgoing arrow; the other side is a incoming solid arrow) attached to each vertex. This is a graph *Γ* drawn on *Σ*. This graph is related to*
(72)$$E_{n,0}\left\{L_{X}\left[tr\left(W_{1}\right)\cdots tr\left(W_{1}\right)\right]\right\}=\hbar^{n}tr\left(W_{1}^{*}\right)\cdots tr\left(W_{V}^{*}\right),$$
*which is (68) for*
$d=1\left(in\;this\;case,\;all\;\Delta^{i}\;has\;weight\;1,\;and\;p_{\left(1\right)}\left(W\right)=tr\left(W\right)\right).\;In\;case\;d>1,\;instead\;of\;each\;tr\left(W_{i}\right)\;we\;have\;the\;product$
$$tr\left({\left(W_{i}\right)}^{\Delta_{1}^{i}}\right)\cdots tr\left({\left(W_{i}\right)}^{\Delta_{\ell_{i}}^{i}}\right)\to tr\left(W_{i}\right).$$

*One may interpret it as a projection of ℓ_i_ polygons to the country (the polygon) labeled by i.*


**Remark** **14.**
*Notice that the answers for the expectation values which were considered above depend only on eigenvalues of $W_{i}^{*}$ or $W_{i}$i=1,…,V.*


## 5. Examples of Matrix Models

Recall that in Section 2.1 we introduced the function $\tau_{r}\left(n,p^{\left(1\right)},p^{\left(2\right)}\right)$. In what follows we use the conventions:
(73)$$\tau_{r}\left(p^{\left(1\right)},p^{\left(2\right)}\right):=\tau_{r}\left(0,p^{\left(1\right)},p^{\left(2\right)}\right)=\sum_{\lambda }r_{\lambda }s_{\lambda }\left(p^{\left(1\right)}\right)s_{\lambda }\left(p^{\left(2\right)}\right),\;r_{\lambda }=r_{\lambda }\left(0\right)$$

It depends on two sets $p^{i}=\left(p_{1}^{\left(i\right)},p_{2}^{\left(i\right)},\ldots \right),\;i=1,2$, and on the choice of an arbitrary function of the variable *r*. (This is an example of the so-called tau function, but we will not use this fact.) As one of their sets, we will choose $p^{2}=p\left(X\right)$ like in (Equation 21), and the second set will be the set of arbitrary parameters. With r=1 we get
(74)$$\tau_{1}\left(p,X\right)=e^{\sum_{m>0}\frac{1}{m}p_{m}\mathrm{tr}\left(X^{m}\right)}$$

For example, if we take
$$r\left(x\right)=\frac{\prod_{i}^{p}\left(a_{i}+x\right)}{\prod_{i}^{q}\left(b_{i}+x\right)},$$
and in addition to $p^{1}=\left(1,0,0,\ldots \right)$ we get the so-called hypergeometric function of the matrix argument:
(75)pFqa1,…,apb1,…,bq|X=∑λdimλ|λ|!sλ(X)∏ip(ai+x)λ∏iq(bi+x)λ

Special cases:
(76)$$e^{\mathrm{tr}X}=\sum_{\lambda }s_{\lambda }\left(X\right)\frac{\dim\;\lambda }{|\lambda |!},\;$$
(77)$$\det{\left(1-zX\right)}^{-a}=\sum_{\lambda }z^{\left|\lambda \right|}{\left(a\right)}_{\lambda }s_{\lambda }\left(X\right)\frac{\dim\;\lambda }{|\lambda |!}$$

Integrals. Using Proposition 2 we obtain

**Theorem** **1.**
*Suppose $W_{1},\ldots ,W_{F}\;and\;W_{1}^{*},\ldots ,W_{V}^{*}$ are dual sets (57). Let sets $p^{i}=\left(p_{1}^{\left(i\right)},p_{2}^{\left(i\right)},p_{3}^{\left(i\right)},\ldots \right),\;i=1,\ldots ,\;\max(F,V)$ be independent complex parameters and r(i),i=1,…, max(F,V) be a set of given functions in one variable.*
(78)$$E_{n_{1},n_{2}}\left\{L_{X}\left[\tau_{r^{\left(1\right)}}\left(p^{1},W_{1}\right)\cdots \tau_{r^{\left(F\right)}}\left(p^{F},W_{F}\right)\right]\right\}$$
(79)$$=\sum_{\lambda }\;r_{\lambda }\;\hbar^{n_{1}\left|\lambda \right|}{\left(\frac{\dim\;\lambda }{|\lambda |!}\right)}^{-n}\prod_{i=1}^{V}s_{\lambda }\left(W_{i}^{*}\right)\prod_{i=1}^{F}s_{\lambda }\left(p^{i}\right),$$
*where each $\tau_{r^{\left(i\right)}}\left(p^{i},W_{i}\right)$ is defined by (21)*
$$r_{\lambda }={\left({\left(N\right)}_{\lambda }\right)}^{-n_{2}}\prod_{i=1}^{F}r_{\lambda }^{\left(i\right)}\left(n\right)$$

*Similarly*
(80)$$E_{n_{1},n_{2}}\left\{L_{X}\left[\tau_{r^{\left(1\right)}}\left(p^{1},W_{1}^{*}\right)\cdots \tau_{r^{\left(V\right)}}\left(p^{V},W_{V}^{*}\right)\right]\right\}$$
(81)$$=\sum_{\lambda }\;r_{\lambda }\;\hbar^{n_{1}\left|\lambda \right|}{\left(\frac{\dim\;\lambda }{|\lambda |!}\right)}^{-n}\prod_{i=1}^{F}s_{\lambda }\left(W_{i}\right)\prod_{i=1}^{V}s_{\lambda }\left(p^{i}\right),$$


**Remark** **15.**
*We recall the convention (Equation 73) In (79) $r_{\lambda }$ is the content product (Equation 16)*
$$r_{\lambda }=r_{\lambda }\left(0\right)=\prod_{(i,j)\in \lambda }r\left(j-i\right)$$
*where*
$$r\left(x\right)={\left(N+x\right)}^{-n_{2}}\prod_{i=1}^{F}r^{\left(i\right)}\left(x\right)$$


To get examples we choose

Dual sets $W_{1},\ldots ,W_{F}↔W_{1}^{*},\ldots ,W_{V}^{*}$;The fraction of unitary matrices given by $n_{2}$;The set of functions $r^{\left(i\right)},\;i=1,\ldots ,F$;The sets $p_{\left(i\right)},\;i=1,\ldots ,F$.

**Remark** **16.**
*Answers in some cases are further simplified. Let us mark two cases*

*(i) Firstly, this is the case when the spectrum of the stars has the form*
(82)$$\mathrm{Spect}\;W_{i}^{*}=\mathrm{Spect}\;I_{N,k_{i}}=\mathrm{diag}\left\{1,1,\ldots ,1,0,0,\ldots ,0\right\},\;i=1,\ldots ,V$$
*where $I_{N,k_{i}}$ is the matrix with $k_{i}$ units of the main diagonal. Such star monodromies obtained in case source matrices have a rank smaller than N. Insertion of such matrices in the left-hand sides of (79) and (81) corresponds to the integration over rectangular random matrices. One should take into account that*
$$s_{\lambda }\left(I_{N,k}\right)={\left(k\right)}_{\lambda }s_{\lambda }\left(p_{\infty }\right),$$
*where we recall the notation*
(83)$${\left(a\right)}_{\lambda }:={\left(a\right)}_{\lambda_{1}}{\left(a-1\right)}_{\lambda_{2}}\cdots {\left(a-\ell +1\right)}_{\lambda_{\ell }},$$

*(ii) The case is the specification of the sets $p^{i}$, i=1,…,F according to the following*


**Lemma** **6.**
*Denote*
(84)$$p_{\infty }=\left(1,0,0,\ldots \right)$$
(85)$$p\left(a\right)=\left(a,a,a,\ldots \right)$$
(86)$$p\left(q,t\right)=\left(p_{1}\left(q,t\right),p_{2}\left(q,t\right),\ldots \right)\;,\;p_{m}\left(q,t\right)=\frac{1-q^{m}}{1-t^{m}}$$

*Then*
(87)$$\frac{s_{\lambda }\left(p\left(a\right)\right)}{s_{\lambda }\left(p_{\infty }\right)}={\left(a\right)}_{\lambda }\;,\;p\left(a\right)=\left(a,a,a,\ldots \right)$$
*where ${\left(a\right)}_{\lambda }:={\left(a\right)}_{\lambda_{1}}{\left(a-1\right)}_{\lambda_{2}}\cdots {\left(a-\ell +1\right)}_{\lambda_{\ell }}$, ${\left(a\right)}_{n}:=a\left(a+1\right)\cdots \left(a+n-1\right)$, where $\lambda =(\lambda_{1},\ldots ,\lambda_{\ell })$ is a partition. More generally*
(88)$$\frac{s_{\lambda }\left(p\left(q,t\right)\right)}{s_{\lambda }\left(p\left(0,t\right)\right)}={\left(q;t\right)}_{\lambda }\;,$$
*where ${\left(q;t\right)}_{\lambda }={\left(q;t\right)}_{\lambda_{1}}{\left({q\;t}^{-1};t\right)}_{\lambda_{2}}\cdots {\left({q\;t}^{1-\ell };t\right)}_{\lambda_{\ell }}$ where ${\left(q;t\right)}_{k}=\left(1-q\right)\left(1-q\;t\right)\cdots \left(1-{q\;t}^{n-1}\right)$ is t-deformed Pochhammer symbol. ${\left(q;t\right)}_{0}=1$ is implied.*

*For such specifications the right-hand side of (79) can ta*

*With such specifications, one can diminish the number of the Schur functions in the right-hand side of (79) (or of (81)) and the right-hand side can take one of the forms:*
(89)$$\sum_{\lambda }r_{\lambda }s_{\lambda }\left(A\right)s_{\lambda }\left(B\right)$$
(90)$$\sum_{\lambda }r_{\lambda }s_{\lambda }\left(A\right)$$
(91)$$\sum_{\lambda }r_{\lambda }$$

*For (Equation 89) there is a determinant representation; for (90) there is a Pfaffian representation and (91) can be rewritten as a sum of products. Indeed if we introduce $r\left(x\right)=e^{T_{x-1}-T_{x}}$, and $\lambda =(\lambda_{1},\ldots ,\lambda_{N})$, then*
(92)$$r_{\lambda }\left(m\right)=\prod_{(i,j)\in \lambda }r\left(m+j-i\right)=e^{T_{m}+\cdots +T_{m-N}}\prod_{i=1}^{N}e^{T_{\lambda_{i}-i+m}}$$

*For instance, one can take r(x)=a+x and get*
(93)$${\left(a\right)}_{\lambda }=\frac{\Gamma \left(a+\lambda_{1}-1\right)\Gamma \left(a+\lambda_{2}-2\right)\cdots \Gamma \left(a+\lambda_{N}-N\right)}{\Gamma (a)\Gamma (a-1)\cdots \Gamma (a-N+1)}$$

*Then we introduce $h_{i}=\lambda_{i}-i+N$ and write*
(94)$$\sum_{\lambda }\frac{{\left(a\right)}_{\lambda }}{{\left(b\right)}_{\lambda }}=\sum_{h_{1}>\cdots >h_{N}\geq 0}\prod_{i=1}^{N}\frac{\Gamma (b-i+1)}{\Gamma (a-i+1)}\frac{\Gamma (h_{i}+a-N)}{\Gamma (h_{i}+b-N)}$$
*where *Γ* is the gamma-function. (In case the argument of gamma-function turns out to be a nonpositive integer one should keep in mind both the enumerator and denominator.) See examples below.*


**Example** **11.**
*See Example 1 and Figure 2a. Take $X_{1}=Z$ and r given by (Equation 13). The example of (79) can be chosen as follows*
(95)E1,0pFqa1,…,apb1,…,bq|ZC1p′Fq′a1′,…,ap′′b1′,…,bq′′|Z†C−1detZZ†α=
(96)p′Fq′a1,…,ap,a1′,…,ap′′,N+αb1,…,bq,b1′,…,bq′′,N|C1C−1,
*corresponding determinantal representation see a (Equation 21).*

*See and Figure 2b which is dual to Figure 2a. An example of (81) can be chosen as*
E1,0p′Fq′a1,…,ap,a1′,…,ap′′,N+αb1,…,bq,b1′,…,bq′′,N|ZC1Z†C−1detZZ†β=
(97)$$\sum_{\lambda }s_{\lambda }\left(C_{1}\right)s_{\lambda }\left(C_{-1}\right)\frac{{\left(N+\alpha \right)}_{\lambda }{\left(N+\beta \right)}_{\lambda })}{{\left({\left(N\right)}_{\lambda }\right)}^{2}}\frac{\prod_{i}^{p}{\left(a_{i}\right)}_{\lambda }}{\prod_{i}^{q}{\left(b_{i}\right)}_{\lambda }}\frac{\prod_{i}^{p^{\prime }}{\left(a_{i}\right)}_{\lambda }}{\prod_{i}^{q^{\prime }}{\left(b_{i}\right)}_{\lambda }},$$

*The determinantal representation of the left-hand side is given by (Equation 15).*


**Example** **12.**
*See Example 2 and Figure 3a. Take $X_{1}=U_{1},\;X_{2}=U_{2}$.*
(98)$$E_{0,2}\left\{e^{\sum_{m>0}\frac{1}{m}p_{m}\mathrm{tr}{\left(U_{1}C_{1}U_{2}C_{2}U_{1}^{†}C_{-1}U_{2}^{†}C_{-2}\right)}^{m}}\right\}=$$
$$=\sum_{\lambda }\frac{1}{{\left({\left(N\right)}_{\lambda }\right)}^{2}}\frac{s_{\lambda }\left(p\right)s_{\lambda }\left(C_{1}C_{-2}C_{-1}C_{2}\right)}{{\left(s_{\lambda }\left(p_{\infty }\right)\right)}^{2}}$$

*Let us take $p=(az,az^{2},az^{3},\ldots )$ and $W_{1}^{*}=I_{N,k}$; see (Equation 85) and (Equation 82). We obtain the left-hand side as*
$$E_{0,2}\left\{\det{\left(1-zU_{1}C_{1}U_{2}C_{2}U_{1}^{†}C_{-1}U_{2}^{†}C_{-2}\right)}^{-a}\right\}=\sum_{\lambda }z^{\left|\lambda \right|}\frac{{\left(a\right)}_{\lambda }{\left(k\right)}_{\lambda }}{{\left({\left(N\right)}_{\lambda }\right)}^{2}}=$$
$$z^{-\frac{1}{2}N\left(N-1\right)}\sum_{h_{1}>\cdots >h_{N}\geq 0}\prod_{i=1}^{N}z^{h_{i}}\frac{{\left(\Gamma (N-i+1)\right)}^{2}}{\Gamma (a-i+1)\Gamma (k-i+1)}\frac{\Gamma \left(h_{i}+a-N\right)\Gamma \left(h_{i}+k-N\right)}{{\left(\Gamma (h_{i})\right)}^{2}}$$
*where $\mathrm{Spect}\;C_{1}C_{-2}C_{-1}C_{2}=\mathrm{Spect}\;I_{N,k}$ see (Equation 94).*


**Example** **13.**
*See Example 3 and Figure 3c.*
$$E_{2,0}\left\{e^{\sum_{m>0}\frac{1}{m}p_{m}^{\left(1\right)}\mathrm{tr}{\left(Z_{1}C_{1}\right)}^{m}+\sum_{m>0}\frac{1}{m}p_{m}^{\left(2\right)}\mathrm{tr}{\left(Z_{2}C_{2}\right)}^{m}}\det{\left(1-zZ_{1}^{†}C_{-1}Z_{2}^{†}C_{-2}\right)}^{-a}\prod_{i=1}^{3}\det{\left(Z_{i}Z_{i}^{†}\right)}^{\alpha }\right\}$$
$$=\sum_{\lambda }z^{\left|\lambda \right|}s_{\lambda }\left(p^{1}\right)s_{\lambda }\left(p^{2}\right){\left(a\right)}_{\lambda }{\left(\frac{{\left(N+\alpha \right)}_{\lambda }}{{\left(N\right)}_{\lambda }}\right)}^{3}$$

*For a determinant representation see (Equation 20)*


**Example** **14.**
*For decoration of Figure 1e we put $X_{i}=Z_{i}$.*
$$E_{2,0}\left\{p_{\Delta }\left(Z_{1}^{†}C_{-1}\right)s_{\lambda }\left(Z_{1}C_{1}Z_{2}C_{2}Z_{2}^{†}C_{-2}\right)\right\}=\delta_{\left|\lambda |,|\Delta \right|}\frac{\dim\;\lambda }{|\lambda |!}\chi_{\lambda }\left(\Delta \right)s_{\lambda }\left(C_{2}\right)s_{\lambda }\left(C_{1}C_{-2}C_{-1}\right)$$
*see (66).*


**Example** **15.**
*Figure 5d in particular yields*
$$E_{2,0}\left\{s_{\lambda }\left(Z_{1}C_{1}Z_{2}C_{2}Z_{3}C_{3}Z_{4}C_{4}\right)s_{\lambda }\left(Z_{3}^{†}C_{-3}Z_{2}^{†}C_{-2}Z_{5}C_{5}\right)s_{\lambda }\left(Z_{5}^{†}C_{-5}Z_{1}^{†}C_{-1}Z_{4}^{†}C_{-4}\right)\right\}=$$
$${\left(\frac{\dim\;\lambda }{|\lambda |!}\right)}^{-5}s_{\lambda }\left(C_{1}C_{-2}C_{-5}\right)s_{\lambda }\left(C_{2}C_{-3}\right)s_{\lambda }\left(C_{5}C_{3}C_{-4}\right)s_{\lambda }\left(C_{4}C_{-1}\right)$$


**Example** **16.**
*In the case below we use an open chain with n edges as in Figure 1b, Figure 2b and Figure 4a.*
(99)$$E_{n,0}\left\{e^{\sum_{m>0}\frac{1}{m}p_{m}^{\left(1\right)}\mathrm{tr}\left({\left(Z_{1}C_{1}Z_{2}C_{2}\cdots Z_{n}C_{n}Z_{n}^{†}C_{-n}\cdots Z_{1}^{†}C_{-1}\right)}^{m}\right)}\right\}=\sum_{\lambda }s_{\lambda }\left(C_{n}\right)s_{\lambda }\left(C_{-1}\right)\frac{s_{\lambda }\left(p\right)}{s_{\lambda }\left(p_{\infty }\right)}\prod_{i=1}^{n-1}\frac{s_{\lambda }\left(C_{i}C_{-i-1}\right)}{s_{\lambda }\left(p_{\infty }\right)}$$

*Graphs dual to the chain look like in Figure 4b.*
$$E_{n,0}\left\{e^{\sum_{m>0}\frac{1}{m}p_{m}\mathrm{tr}\left({\left(Z_{1}^{†}C_{-1}\right)}^{m}\right)+\sum_{m>0}\frac{1}{m}p_{m}^{\left(n\right)}\mathrm{tr}\left({\left(Z_{n}C_{n}\right)}^{m}\right)}\prod_{i=1}^{n-1}e^{\sum_{m>0}\frac{1}{m}p_{m}^{\left(i\right)}\mathrm{tr}\left({\left(Z_{i}C_{i}Z_{i+1}^{†}C_{-i-1}\right)}^{m}\right)}\right\}$$
$$=\sum_{\lambda }s_{\lambda }\left(p\right)s_{\lambda }\left(C_{1}C_{2}\cdots C_{n}C_{-n}\cdots C_{-1}\right)\prod_{i=1}^{n}\frac{s_{\lambda }\left(p^{i}\right)}{s_{\lambda }\left(p_{\infty }\right)}$$

*These relations generalize product (Equation 44) and (Equation 45).*


**Example** **17.**
*Our graph is a polygon with n edges and n vertices (stars); see Figure 1d, Figure 2a and Figure 4c for examples.*
(100)$$E_{n,0}\left\{e^{\sum_{m>0}\frac{1}{m}p_{m}^{\left(1\right)}\mathrm{tr}{\left(Z_{1}C_{1}Z_{2}C_{2}\cdots Z_{n}C_{n}\right)}^{m}+\sum_{m>0}\frac{1}{m}p_{m}^{\left(2\right)}\mathrm{tr}{\left(Z_{n}^{†}C_{-n}Z_{n-1}^{†}C_{1-n}\cdots Z_{1}^{†}C_{-1}\right)}^{m}}\prod_{i=1}^{n}\det{\left(Z_{i}Z_{i}^{†}\right)}^{\alpha }\right\}$$
$$=\sum_{\lambda }s_{\lambda }\left(p^{1}\right)s_{\lambda }\left(p^{2}\right)\prod_{i=1}^{n}\frac{{\left(N+\alpha \right)}_{\lambda }s_{\lambda }\left(C_{i}C_{-i-1}\right)}{{\left(N\right)}_{\lambda }s_{\lambda }\left(p_{\infty }\right)}$$
*(where we put $C_{-n-1}=C_{-1}$).*

*A graph dual to the polygon can be viewed as two-stars graph with n edges which connect stars; see Figure 1d and Figure 4d as examples.*
(101)$$E_{n,0}\left\{\prod_{i=1}^{n}e^{\sum_{m>0}\frac{1}{m}p_{m}^{\left(i\right)}\mathrm{tr}{\left(Z_{i}C_{i}Z_{i+1}^{†}C_{-i-1}\right)}^{m}}\right\}=\sum_{\lambda }s_{\lambda }\left(C_{1}C_{2}\cdots C_{n}\right)s_{\lambda }\left(C_{-n}C_{1-n}\cdots C_{-1}\right)\prod_{i=1}^{n}\frac{s_{\lambda }\left(p^{i}\right)}{s_{\lambda }\left(p_{\infty }\right)}$$

*To apply determinantal formulas one should use Remark 16.*


**Example** **18.**
*Consider the star-graph with n-rays which end at other stars (see Figure 5a where n=3). This situation corresponds to (Equation 43).*
(102)$$E_{n,0}\left\{e^{\sum_{m>0}\frac{1}{m}p_{m}\mathrm{tr}\left(Z_{1}C_{1}Z_{1}^{†}C_{-1}\cdots Z_{n}C_{n}Z_{n}^{†}C_{-n}\right)}\prod_{i=1}^{n}\det{\left(Z_{i}Z_{i}^{†}\right)}^{\alpha }\right\}=$$
$$\sum_{\lambda }s_{\lambda }\left(p\right)s_{\lambda }\left(C_{-1}C_{-2}\cdots C_{-n}\right)\prod_{i=1}^{n}\frac{{\left(N+\alpha \right)}_{\lambda }s_{\lambda }\left(C_{i}\right)}{{\left(N\right)}_{\lambda }s_{\lambda }\left(p_{\infty }\right)}$$

*A similar model was studied in [16,22]. It has the determinantal representation (Equation 21) in case all $W_{i}^{*}$ except one are of form (Equation 82). There is the determinantal representation (Equation 15) in case we specialize the set p according to Lemma 6 and choose each $W_{i}^{*}$ except two be in form (Equation 82).*

*Now, let us choose the dual graph (this is petel graph. (see Figure 5b where n=3)) and consider*
(103)$$E_{n,0}\left\{\det{\left(1-zZ_{1}^{†}C_{-1}Z_{2}^{†}C_{-2}\cdots Z_{n}^{†}C_{-n}\right)}^{-a}\prod_{i=1}^{n}\det{\left(Z_{i}Z_{i}^{†}\right)}^{\alpha }e^{\sum_{m>0}\frac{1}{m}p_{m}^{\left(i\right)}\mathrm{tr}\left({\left(Z_{i}C_{i}\right)}^{m}\right)}\right\}=$$
$$\sum_{\lambda }z^{\left|\lambda \right|}{\left(a\right)}_{\lambda }s_{\lambda }\left(C_{1}C_{-1}C_{2}C_{-2}\cdots C_{n}C_{-n}\right)\prod_{i=1}^{n}\frac{{\left(N+\alpha \right)}_{\lambda }s_{\lambda }\left(p^{i}\right)}{{\left(N\right)}_{\lambda }s_{\lambda }\left(p_{\infty }\right)}$$

*By Remark 16 we find all cases where the determinantal representations (Equation 20) or (Equation 21) exist.*


**Remark** **17.**
*Notice the following symmetry: the left-hand side produces the same right-hand side if we permute the set of exponents $\alpha_{1},\ldots ,\alpha_{n},a-N$.*


**Example** **19.**
*Below g=1,2,…. (For the case g=1 see Figure 1a and zoomed Figure 3a; for g=2 see Figure 5e).*
$$E_{0,2g}\left\{e^{\sum_{m>0}\frac{1}{m}p_{m}\mathrm{tr}{\left(U_{a_{1}}C_{a_{1}}U_{b_{1}}C_{b_{1}}U_{a_{1}}^{†}C_{-a_{1}}U_{b_{1}}^{†}C_{-b_{1}}\cdots U_{a_{g}}C_{a_{g}}U_{b_{g}}C_{b_{g}}U_{a_{g}}^{†}C_{-a_{g}}U_{b_{g}}^{†}C_{-b_{g}}\right)}^{m}}\right\}$$
$$=\sum_{\lambda }{\left(\frac{\dim\;\lambda }{|\lambda |!}\right)}^{-2g}s_{\lambda }\left(W^{*}\right)s_{\lambda }\left(p\right)$$
*where*
$$W^{*}=C_{-a_{1}}C_{b_{1}}C_{a_{1}}C_{-b_{1}}\cdots C_{-a_{g}}C_{b_{g}}C_{a_{g}}C_{-b_{g}}$$

*In particular, if $\mathrm{Spect}\;W^{*}=I_{N,k}$, then*
$$E_{0,2g}\left\{\det{\left(I_{N}-zU_{a_{1}}C_{a_{1}}U_{b_{1}}C_{b_{1}}U_{a_{1}}^{†}C_{-a_{1}}U_{b_{1}}^{†}C_{-b_{1}}\cdots U_{a_{g}}C_{a_{g}}U_{b_{g}}C_{b_{g}}U_{a_{g}}^{†}C_{-a_{g}}U_{b_{g}}^{†}C_{-b_{g}}\right)}^{-a}\right\}$$
(104)$$=\sum_{\lambda }z^{\left|\lambda \right|}{\left(\frac{\dim\;\lambda }{|\lambda |!}\right)}^{2-2g}\frac{{\left(a\right)}_{\lambda }{\left(k\right)}_{\lambda }}{{\left({\left(N\right)}_{\lambda }\right)}^{2g}}$$

*Taking into account that*
$$\dim\;\lambda =\frac{\prod_{i<j}\left(h_{i}-h_{j}\right)}{\prod_{i=1}^{N}\Gamma \left(h_{i}+1\right)}$$
*we can interpret that (Equation 104) is a discrete beta-ensemble where β=2−2g.*


Exotic models. An example. There are some more tricky problems which can be solved which can be solved in steps. Let me consider the simplest example. Look at Lemma 3. Suppose *A* and *B* depend in any way on an additional matrix $Z_{1}$; in any case, however, their product has a familiar form:
(105)$$A=A\left(Z_{1},Z_{1}^{†}\right),\;B=B\left(Z_{1},Z_{1}^{†}\right),\;AB=Z_{1}C_{1}Z_{1}^{†}C_{-1}$$

Say, $A=Z_{1}^{a}e^{-Z_{1}^{†}},\;B=e^{Z_{1}^{†}}Z_{1}^{†}Z_{1}^{1-a}$ which looks horrible. However, applying sequentially the series ${{}_{0}F}_{0}$, then (Equation 37), where $Z=Z_{1}$, and then (Equation 31), where $Z=Z_{2}$ one obtains
(106)$$E_{2,0}\left\{e^{\mathrm{tr}\left(Z_{1}Z_{2}^{a}e^{-Z_{2}^{†}}\right)+\mathrm{tr}\left(Z_{1}^{†}e^{Z_{2}^{†}}Z_{2}^{†}Z_{2}^{1-a}\right)}\right\}=\sum_{d\geq 0}\sum_{\lambda \in Y_{d}}s_{\lambda }\left(I_{N}\right)s_{\lambda }\left(C\right)=\det{\left(I_{N}-C\right)}^{-1}$$

It will be interesting to do the same with other ensembles of random matrices; Ginibre ensembles of real and quaternionic matrices; and ensembles of Hermitian matrices: complex, real and quaternionic.

## 6. Discussion

In this article, we examined matrix integrals of a certain type. We called them matrix models associated with children’s drawings—the so-called dessin d’enfants. They include some well-known models that have found applications in the theory of information transfer and the theory of quantum chaos. We hope that our matrix integrals will be in demand. We think that these models are related to quantum integrable systems [38,39,40], but this topic is waiting for its development; we expect connections with [41,42,43,44,45,46,47,48,49].

## Figures and Tables

**Figure 1 entropy-22-00972-f001:**
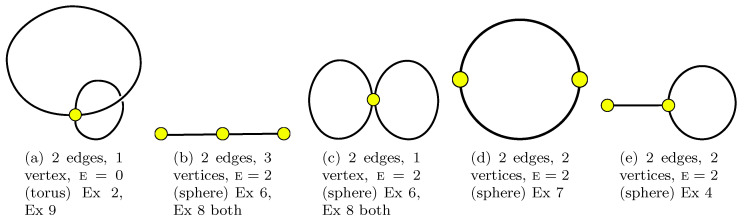
All possible graphs with 2 edges (two matrix models). Graph (**b**) is dual to (**c**). Ex means an example below.

**Figure 2 entropy-22-00972-f002:**
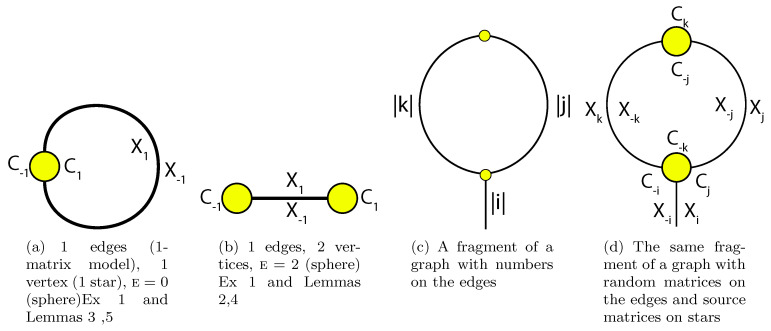
Decorated graphs.

**Figure 3 entropy-22-00972-f003:**
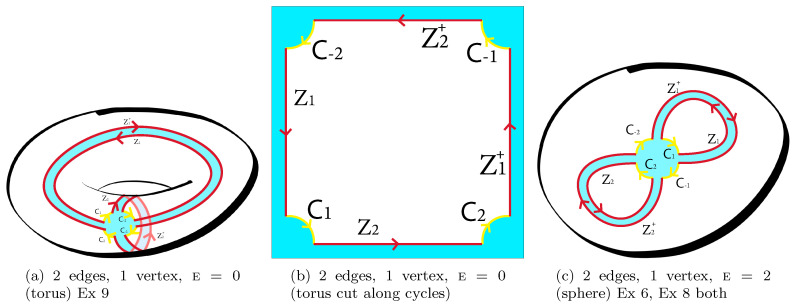
Graphs with 2 edges (two matrix models) where (**a**) is the zoom of (a) in Figure 1 and (**c**) is the zoom of (c) in Figure 1.

**Figure 4 entropy-22-00972-f004:**
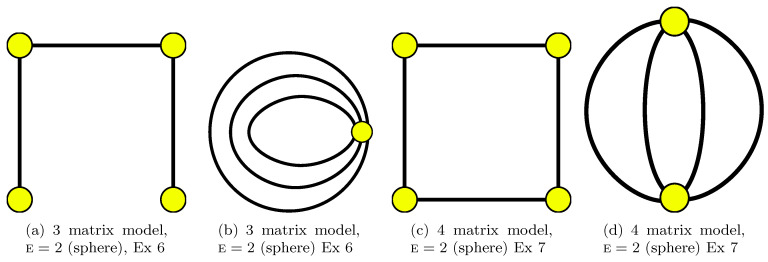
Graphs drawn without decoration. Graph (**a**) is dual to (**b**). Graph (**c**) is dual to (**d**).

**Figure 5 entropy-22-00972-f005:**
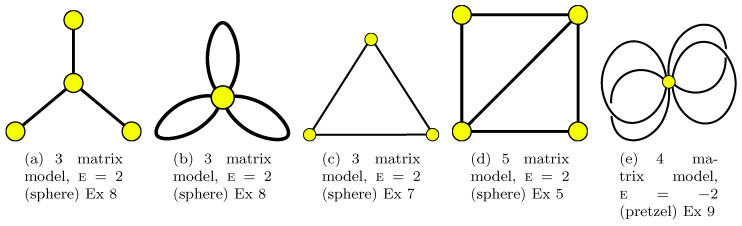
Graphs drawn without decoration. Graph (**a**) is dual to (**b**).

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
