# Peer review of "On Products of Random Matrices"

_entropy, 2020, doi:10.3390/e22090972_

Round 1

Reviewer 1 Report

The manuscript is devoted to a family of models involving random matrices. In particular, the Authors discuss a special kind of graphs referred to as ``dessins d’enfant .’’ Such graphs’ edges are associated with random matrices. Presented models are a generalization of those already discussed in the literature. They are based on the Schur functions and application of the so-called Hurwitz numbers. The Authors present both algebraical and geometrical views.

The article is well written in general. It starts from the introductory part in which the Authors have thoroughly explain all ideas which are necessary for understanding the merit of the paper. The proposed ideas are illustrated by some examples, showing how they are working. Moreover, some more detailed calculations/considerations are presented in the appendix. The Authors also give a comprehensive list of references that can be helpful for the Readers in their further studies. 

However, there is one doubt concerning the article. Namely, the Special Issue is related to the devoted to the topics related to the quantum information. Therefore, I feel lack of, at least one,  an example showing how the ideas presented in the manuscript could be applied in the description of quantum systems (for instance, in the detection of the entanglement).

In my opinion, the article can be accepted for publication, but after improving it according to the mentioned above point. My suggestion is the ``minor revision''.

Author Response

Dear reviewer,

We thank you for thorough reading of the manuscript and valuable comments on it.

Actually the investigation of connection of the advised family of matrix models with the theory of quantum information seems very promissing. We added some valuable for us referencies on the subject. But we found impossible of further development of this subject in the manuscript due to restrictions on the release date of this special issue.

With best regards,

Dmitry Vasiliev

Reviewer 2 Report

This is a very nice article.  Matrix models associated
with children’s drawings - the so-called dessins d’enfants. They include some well-known models that have found application in the theory of information transfer and the theory of quantum chaos.  

The paper contains many well-studied and thoughtful  examples. The paper will of interest to researchers in general, in particular, researchers in combinatorial theory .

The results have merit and they should be published. 

Author Response

Dear reviewer,

We thank you for thorough reading of the manuscript and comments on it.

With best regards,

Dmitry Vasiliev

Reviewer 3 Report

The authors introduce and study a very geneal class of random matrices, obtained from a (not necessarily planar) combinatorial map by attaching to the edges of the maps independent Gaussian or Haar-unitary random matrices. Such models are interesting from several perspective: the generality given by the geometrical properties of the combinatorial map allows to obtain several types of products of random matrices of interest, while mixing Gaussian and random unitary matrices has its own, separate, interest.

Once the model is introduced and described from two perspectives, the authors compute several integrals with respect to these random models. The integrals involve products of several classes of symmetric polynomials, and several very interesting connections to problems in combinatorics are obtained.

The results presented are new, and interesting to at least two communities: random matrix theory and combinatorics. The proofs are correct, and the paper is globally well-written, although I think that the authors could have expended more effort into the presentation of the results. This is the main criticism I have towards this paper. The main results, such as Propositions 1,2, and Theorem 1 could have been stated in a clearer manner (see the points below). Moreover, feel that the presentation could have been improved by using a more standard "statement -> proof" way of writing the paper. For instance, nowhere in the paper is mentioned how one does actually compute Gaussian or unitary integrals, which is the main topic of the paper: going from expressions of the form E[ function of L_X(W)'s] to expressions of the form [function of W^*'s].

I consider that the results of the paper are interesting enough to deserve publication in "Entropy", but I would like to see these results better presented and appropriate proofs given.

Below is a list of small typos and remarks that the authors might want to consider when preparing a revision:

- page 2, just before Section 2: "in the form of determinants or in the form of works." the last words do not make sense
- page 2, remark 1 (and also later): there is an asterisk after "Remark 1."
- page 6, section 2.2: "is called n independent complex Ginibre ensembles" the "n" should read "n_1"
- page 6, formula (31): this is an important example, since it corresponds to mixed moments of Gaussian matrices. Some precise computations, say for p_(2,0) and p_(1,1) could be useful here
- page 6, eq (30) and page 7, eq (36): one could recall here Eq. (8), where the explicit formula for s_lambda(p_infty) is given
- page 7, top: there is a dot after the "Proof" environment
- page 8, Lemma 4: could the authors comment on a possible connection to the Harish Chandra-Itzykson-Zuber (HCIZ) integral here?
- page 9, Section 3.2 top: "the edges of the graph in figure () do not intersect" the figure number is missing
- page 9: "Such a graph is sometimes called a (clean) dessins d’enfants" the authors should use here (and elsewhere) the singular of the French word "dessin"
- page 10, center: the sentence "We attribute the number −i" is out of place here
- page 11, top: "LX(W_−1) = X_−1C_=1" the subscript of C should be minus, not equal
- page 11: in the statement of Proposition 1, the authors could recall the meaning of the parameters n_1, n_2, and hbar
- page 11/12: the authors could mention that in Proposition 1, each matrix Z / Z^* appears only once, hence the integrand is multi-linear
- page 13, "Cut-or-join procedure" paragraph: there is an "dots" out of place
- page 16, and later also: the expression "Let me give an idea." is used, which is too informal for a scientific paper
- page 17, after eq. (73): there is an extra dot at the begging of the linear
- page 18, Remark 13: "Important notice" is not well suited here
- page 21, paragraph "Exotic models. An example.": there are many daggers which are not in the right place, and there is a typo in "ontains"

Author Response

Dear reviewer,

We thank you for thorough reading of the manuscript and valuable comments on it. 

We ubdated the paper acording all the list of your remarks. Please find attached an updated version of our paper.

With best regards,

Dmitry Vasiliev
